# Photooxidation triggered ultralong afterglow in carbon nanodots

Guang-Song Zheng[1,5], Cheng-Long Shen [1,5], Chun-Yao Niu[1,5], Qing Lou [1 ✉], Tian-Ci Jiang[2,3], Peng-Fei Li[2,3], Xiao-Jing Shi [4], Run-Wei Song[1], Yuan Deng[1], Chao-Fan Lv[1], Kai-Kai Liu[1], Jin-Hao Zang[1], Zhe Cheng[2,3], Lin Dong [1] & Chong-Xin Shan [1 ✉]

It remains a challenge to obtain biocompatible afterglow materials with long emission wavelengths, durable lifetimes, and good water solubility. Herein we develop a photooxidation strategy to construct near-infrared afterglow carbon nanodots with an extra-long lifetime of up to 5.9 h, comparable to that of the well-known rare-earth or organic long-persistent luminescent materials. Intriguingly, size-dependent afterglow lifetime evolution from 3.4 to 5.9 h has been observed from the carbon nanodots systems in aqueous solution. With structural/ultrafast dynamics analysis and density functional theory simulations, we reveal that the persistent luminescence in carbon nanodots is activated by a photooxidation-induced dioxetane intermediate, which can slowly release and convert energy into luminous emission via the steric hindrance effect of nanoparticles. With the persistent near-infrared luminescence, tissue penetration depth of 20 mm can be achieved. Thanks to the high signal-to-background ratio, biological safety and cancer-specific targeting ability of carbon nanodots, ultralong-afterglow guided surgery has been successfully performed on mice model to remove tumor tissues accurately, demonstrating potential clinical applications. These results may facilitate the development of long-lasting luminescent materials for precision tumor resection.

Afterglow, also known as persistent luminescence, is a long-lifetime emission[1-3]. The favorable luminescent ability endows afterglow materials with application potential in data storage, optoelectronic devices, chemical sensors, and bioimaging[4-7]. Especially in bioimaging techniques, afterglow bioimaging with long-wavelength emission has excellent signal-to-background ratio and large tissue penetration depth since there is no excitation light and tissue autofluorescence interference[8-10]. Recently, metal-free organics and rare-earth-doped inorganic crystals have been observed with long-term afterglow emission. For organic afterglow phosphors, afterglow emission involves the spin-flip process in the electron transitions between the excited singlet and triplet states through the intersystem or even reverse intersystem crossing, resulting in relatively long-lifetime room temperature phosphorescence or thermally activated delayed fluorescence[11-13]. Nevertheless, their afterglow durations are always limited in the range from microsecond to several seconds[14-16]. In addition, the triplet excitons from organics are usually easily quenched by the dissolved oxygen in the physiological water environment, and low photostability is another difficult problem for these afterglow materials[17,18]. By contrast, inorganic afterglow systems are free of

[1]Henan Key Laboratory of Diamond Optoelectronic Materials and Devices, Key Laboratory of Material Physics, Ministry of Education, and School of Physics and Microelectronics, Zhengzhou University, Zhengzhou 450052, China. [2]Department of Respiratory and Critical Care Medicine, The First Affiliated Hospital of Zhengzhou University, Zhengzhou 450052, China. [3]Henan Key Laboratory for Pharmacology of Liver Diseases, Institute of Medical and Pharmaceutical Sciences, Zhengzhou University, Zhengzhou 450052, China. [4]Academy of Medical Sciences, Zhengzhou University, Zhengzhou 450052, China. [5]These authors contributed equally: Guang-Song Zheng, Cheng-Long Shen, Chun-Yao Niu. ✉e-mail: louqing1986@zzu.edu.cn; cxshan@zzu.edu.cn

photobleaching, environmentally insensitive, and easy to obtain long persistent luminescence, deriving from multiple trapping-detrapping process of charge carriers in the native defects of inorganic matrixes because of the introduction of rare-earth or nonnoble metal ions like $Eu^{2+}$, $Zn^{2+}$, $Mn^{2+}$, etc.[19–21]. However, diversified limitations, such as tedious preparation, potential cytotoxicity, low solubility, and high cost, seriously hinder their practical application prospect in all kinds of bioimaging fields, such as imaging-guided surgery, early diagnosis of inflammation and ultrasensitive detection of cancer cells. Hence, developing long persistent afterglow materials and gaining deep insight into their luminescent features or mechanisms are of great importance.

Carbon nanodots (CDs), as an emerging phosphor of carbon-based luminescent nanomaterials with tunable wavelength, high physicochemical inertness, low biological toxicity and easy for preparation, has aroused extensive research interests in the fields of biological imaging, information encryption, light-emitting devices and so on[22–26]. With $sp^2/sp^3$ hybrid conjugated structures and abundant heteroatom dopants, CDs can be designed to achieve afterglow emission via embedding into diverse matrices like poly(vinyl alcohol), sodium hydroxide, cyanuric acid, and urea, which can stabilize the triplet excited states and suppress non-radiative transition through the fixation of hydrogen bonds, covalent bonds or ionic bonds[27–30]. However, most of CD-based afterglow emissions are located in blue or green region, while the reports on near-infrared afterglow CDs are still rare. Additionally, the afterglow of CDs usually suffers from drastic quenching in physiological water environment. To date, although red and near-infrared (NIR) aqueous-related afterglow has been observed through forming covalent bonds between CDs and colloidal silica or constructing special p–n junction in carbon-based structures, their afterglow lifetimes are all shorter than one second[31,32]. Such a short duration of afterglow CDs in solution present adverse restrictions to their applications in the field of biological imaging severely. In such a scenario, the development of NIR afterglow CDs with ultralong lifetime in aqueous solution remains a formidable challenge.

Owing to the aggressive proliferation of cancer cells and insufficient blood supply in tumors, the tumor environment features angiogenesis, maladjusted biosynthesis intermediates, acidosis, and hypoxia[33,34]. Actually, hypoxia is an important character of malignant tumors, which depends on tumor angiogenesis and rapid growth of tumor cells. The newly formed blood vessels are different from normal blood vessels, and the poor oxygen supply capacity and slow blood flow will lead to the insufficient oxygen and nutrients transported in tumor cells, resulting in an imbalance in oxygen supply and consumption in tumor tissue[35,36]. In addition, due to the abnormal metabolism of tumor cells, the tumor environment extensively presents excess reactive oxygen species (ROS) of $H_2O_2$, $\bullet O^{2-}$ and $\bullet OH$[37,38], and the superfluous energy transfer between ROS and triplet excitons will seriously limit the bioimaging assisted by phosphorescence or thermal activation delayed fluorescence. Therefore, it is still a huge challenging issue to achieve the precise diagnosis and therapy of tumor under such complex oxygen environment. A facile strategy to achieve ultralong-persistent and high-quality afterglow imaging in intricate biological environment is using CDs as emitter via storing the excited energy in long-lifetime intermediate states or defects, and then tardily emitting photons after the removal of light irradiation (Fig. 1). With special carbon conjugated structure, CDs can act well as both electron donors and acceptors. Under irradiation from high-energy photons, the ground-state electrons of CDs can transit to excited state and then the excited electrons can leap to ground state via the radiative recombination or the non-radiation transition, like the electron exchange with dissolved oxygen to form singlet oxygen ($^1O_2$). Due to the appropriate oxidation of $^1O_2$, the CDs can be oxidized and simultaneously produce one kind of energetic intermediates. The metastable intermediates can further slowly decompose and transfer the chemical energy to the CDs,

resulting in the excitation and ulteriorly long-persistent afterglow. With the photooxidation-assisted afterglow, the issue about the dissolved oxygen-caused quenching of triplet excitons in common phosphorescence or thermal activation delayed fluorescence can be effectively prevented. In addition, compared with traditional organic small molecule afterglow materials, the high polymerization and large size of CDs may inactivate and slow down the photooxidation process, further boosting long persistent of afterglow emission. Therefore, it is possible to obtain ultralong lifetime NIR emissive CDs by using the photooxidation strategy.

In this work, to validate the above hypothesis, we chose biomass as precursors to design and synthesize NIR emissive CDs. The afterglow lifetime of the CDs demonstrates size-dependent features, namely the duration can increase to 156% of the original value with increasing their size from 1.9 to 5.8 nm. After modification with amphiphilic polymer, the CDs perform good water solubility and prolonged lifetime of 46.4 min up to 5.9 h, which outperforms most of super-long persistent luminescent materials. The changes in chemical structural and ultrafast dynamic analysis after light irradiation as well as density functional theory (DFT) calculation confirm that oxidative $^1O_2$ is generated by photosensitization to cause the generation of energy-rich intermediates, which can act as 'energy storage batteries' to slowly release energy to CDs and induce their persistent afterglow. In addition, the conjugated $sp^2$ hybrid structure of CDs with elevated size also has a great steric hindrance effect and few active sites favoring photooxidation, which can further retard the luminescent process and finally achieve ultralong lifetimes in CDs. Besides, CDs present good bio-safety based on elaborate toxicity evaluation large tissue penetration depth of 20 mm and favorable tumor enrichment ability. As proof of principle, NIR afterglow CDs have been displayed to be suitable for use in imaging tumors and guidance for precise surgical resection in mice model, paving the way for the development of application potential of carbon-based nanomaterials in imaging-guided surgery with afterglow bioimaging technique.

## Results

### Synthesis of near-infrared ultralong afterglow CDs

In this study, CDs with NIR emission characteristics were prepared by a solvent thermal method with *Solanum nigrum L* as precursor. As shown in Fig. 2a, the CDs were prepared through solvothermal treatment of *Solanum nigrum L* in a solution of ethanol, then the crude products were purified via dry silica column chromatography using the dichloromethane as the solvent under reduced pressure, and the production yields of CDs is about 1.35% (Supplementary Note 1 and Supplementary Fig. 1), and the production yields of CDs can be further improved by expanding the reaction chamber (Supplementary Note 2 and Supplementary Fig. 2). For comparison, CDs in three sizes were prepared with changing the reaction temperatures. As illustrated in Fig. 2b, all the obtained CDs have obvious fluorescence and afterglow emission with an excellent repeatability (Supplementary Note 3, Supplementary Figs. 3–6). Then, the morphology of CDs-1, CDs-2, and CDs-3 has been investigated via transmission electron microscopy (TEM) (Fig. 2c, h, m). These TEM images illustrate a decrease in size distribution from 5.8 to 1.9 nm, and the high-resolution TEM (HR-TEM) images of these CDs illustrate a well-resolved lattice spacing of 0.21 nm (Supplementary Fig. 7), corresponding to the (100) interplanar spacing of graphitic carbon structure. Atomic Force Microscopy (AFM) images have been performed on the CDs-1, CDs-2, and CDs-3, and the results reveal the uniform heights of 8.6, 4.4, and 2.9 nm for the CDs-1, CDs-2 and CDs-3 (Supplementary Fig. 8), which are consistent with the size variations observed in the TEM images. Meanwhile, the X-ray diffraction (XRD) patterns of the CDs prove a similar peak at around 23°, which can be attributed to the graphite (002) lattice spacing (Supplementary Fig. 9). X-ray photoelectron spectroscopy (XPS) and Fourier transform infrared (FT-IR) spectra are employed to investigate

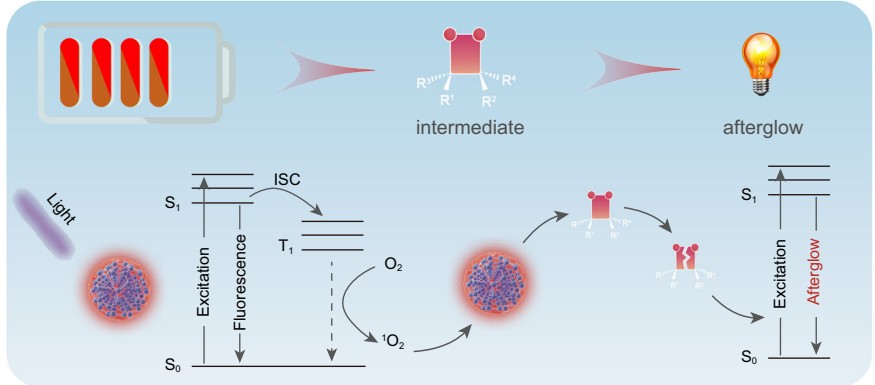

**Fig. 1 | Photooxidation-a new strategy for ultralong afterglow in carbon nanodots.** Schematic illustration of long-persistent afterglow triggered by photooxidative reaction.

the surface functional groups of the CDs. The full-survey XPS spectra illustrate that all the CDs exhibit the composition of C (284.6 eV), N (399.6 eV) and O (532.1 eV) elements, and have a similar elemental content (Fig. 2d, i, n). Besides, the high-resolution XPS C1s spectra of the CDs can be deconvoluted into three Gaussian peaks, corresponding to $sp^2$ C (C−C/C=C) at 284.5 eV, $sp^3$ C (C−N) at 285.3 eV, (C=O)−O at 288.4 eV and C=O at 289.3 eV. And all these CDs demonstrate a highly controllable $sp^2$ hybrid (the ratio of $sp^2$/$sp^3$ C varies from 78.3% to 86.9%), indicating a good foundation for CDs to form 'energy storage batteries'. As depicted in Supplementary Fig. 10, the FT-IR spectra of these CDs are similar with the stretching vibration of C−N (1350 $cm^{-1}$), C=C (1600 $cm^{-1}$), O−H (3200 $cm^{-1}$), and N−H (3140 $cm^{-1}$), which are consistent with the XPS results of CDs. The both XPS and FT-IR characterization verifies that these CDs have similar $sp^2$ carbon-core structures with abundant N/O functional groups on their surface.

Then, the photophysical properties of these CDs have been further investigated. As presented in Fig. 2e, j, o, all the fluorescence peaks appear at around 670 and 720 nm under the excitation of 410 nm light. With the directly integrating sphere measurement, the PL QYs of CDs-1, CDs-2 and CDs-3 are calculated as 28.7%, 31.8% and 31.0%, respectively (Supplementary Fig. 11). The time-resolved spectra of the CDs with the 670 and 720 nm can be well fitted to a single exponential decay with the lifetime of 5.36 and 5.43 ns (Supplementary Fig. 12), suggesting distinct emission sources of the two emission centers. The emission at 670 nm may originate from the band-edge recombination of CDs and the fluorescence at 720 nm may come from the surface-localized excitonic vibrational fine emission bands in the CDs[39–42]. (Supplementary Note 4, Supplementary Figs. 13–15). Moreover, the UV-vis absorption reveals that all the CDs have very broad absorption ranging from UV to deep-red waveband (Supplementary Fig. 16). Meanwhile, obvious afterglow with the wavelength extending to the NIR region can be observed from the CDs, whose afterglow spectra are consistent with their fluorescence spectra (Fig. 2f, k, p). The wavelength of its afterglow aligns with that of fluorescence emission (Supplementary Fig. 17), providing evidence that it originates from the same emission level. This is different from the traditional phosphorescence emission behavior, in which the afterglow wavelength is usually longer than their fluorescence due to the lower triplet-state energy level in comparison with their singlet state[43,44]. In addition, the dynamic luminescence of these CDs has been tested. All the afterglow emission time of these CDs in ethanol solution is over 4 h, and the corresponding lifetimes are fitted to 46.4, 42.5 and 29.7 min for CDs-1, CDs-2, and CDs-3 (Fig. 2g, l, q), exhibiting a distinct size-dependent afterglow lifetime growth characteristics with good repeatability for the CDs (Supplementary Note 5, Supplementary Figs. 18−21). Furthermore, the effective pre-irradiation wavelength of afterglow emission is in accordance with the absorption spectrum of the CDs, ranging from UV to deep-red

region and even white light (Supplementary Fig. 22). The excellent deep-red to NIR afterglow emission and wide range excitation of CDs demonstrate their broad application prospects.

## Mechanism of the afterglow emission

The luminescence mechanism of the long-persistent afterglow properties has been further probed using CDs-1 as a representative. Firstly, the structure and optical properties of the CDs before and after light irradiation were investigated in detail. As shown in Fig. 3a, it can be clearly observed that the CDs undergo a solubility transition from good dissolution in nonpolar dichloromethane to high dispersibility in polar water prior to and after light irradiation, indicating that there is a change from fat solubility to water solubility for the CDs after light irradiation. The morphologies of CDs after irradiation were studied via TEM analysis (Fig. 3b). CDs still maintain their shape after irradiation, manifesting that changes caused by light irradiation only occur on the chemical composition and structure of CDs. To further explore their composition and structural evolution before and after light irradiation, the nuclear magnetic resonance (NMR), FT-IR, and XPS have been measured. The $^1H$ NMR spectra of the CDs before and after irradiation display the obvious signals change of functional groups, namely, the signal of −OCOCH$_3$ at 4.8 ppm is enhanced after optical excitation, implying the increase of oxygen-containing bonds on the surface of CDs after irradiation (Fig. 3c)[22]. As illustrated in Fig. 3d, the FT-IR spectra also reveal the stretching vibration of C−N (1350 $cm^{-1}$), C=C (1600 $cm^{-1}$), O−H (3200 $cm^{-1}$) after the light irradiation. Meanwhile, the peak position of the stretching vibration in CDs does not change obviously, indicating that the bond types on the surface of the CDs are unchanged. Nevertheless, the intensity of C=O−OH related stretching vibration is significantly enhanced, proclaiming that the carboxyl on the surface of CDs increases after the light irradiation. The full-survey XPS spectra illustrate that the elements of C (284.6 eV), N (399.6 eV) and O (532.1 eV) still exist in the CDs after the light irradiation, being accompanied by the elevated oxygen content in the CDs from 9.2% to 14.7% (Supplementary Fig. 23). The high-resolution XPS spectra of O1s for the CDs could be deconvolved into three Gaussian peaks corresponding to C=O, C−O/O−H and O=C−O, respectively (Fig. 3e, f), whose O=C−O content increased remarkably with the light irradiation, which is consistent with the result of FT-IR spectra. The high-resolution XPS spectra of N1s for the CDs, the content and type of N element did not change significantly after light irradiation (Supplementary Note 6 and Supplementary Fig. 24), indicating that oxidation primarily occurred on the surface of the CDs. In addition, a new strong signal appears at m/z of 391 in the mass spectrum of the CDs after light-irradiation treatment, hinting the emergence of new products probably associated with after light irradiation (Supplementary Fig. 25). Hence, the photochemical structural transformation arises in the CDs

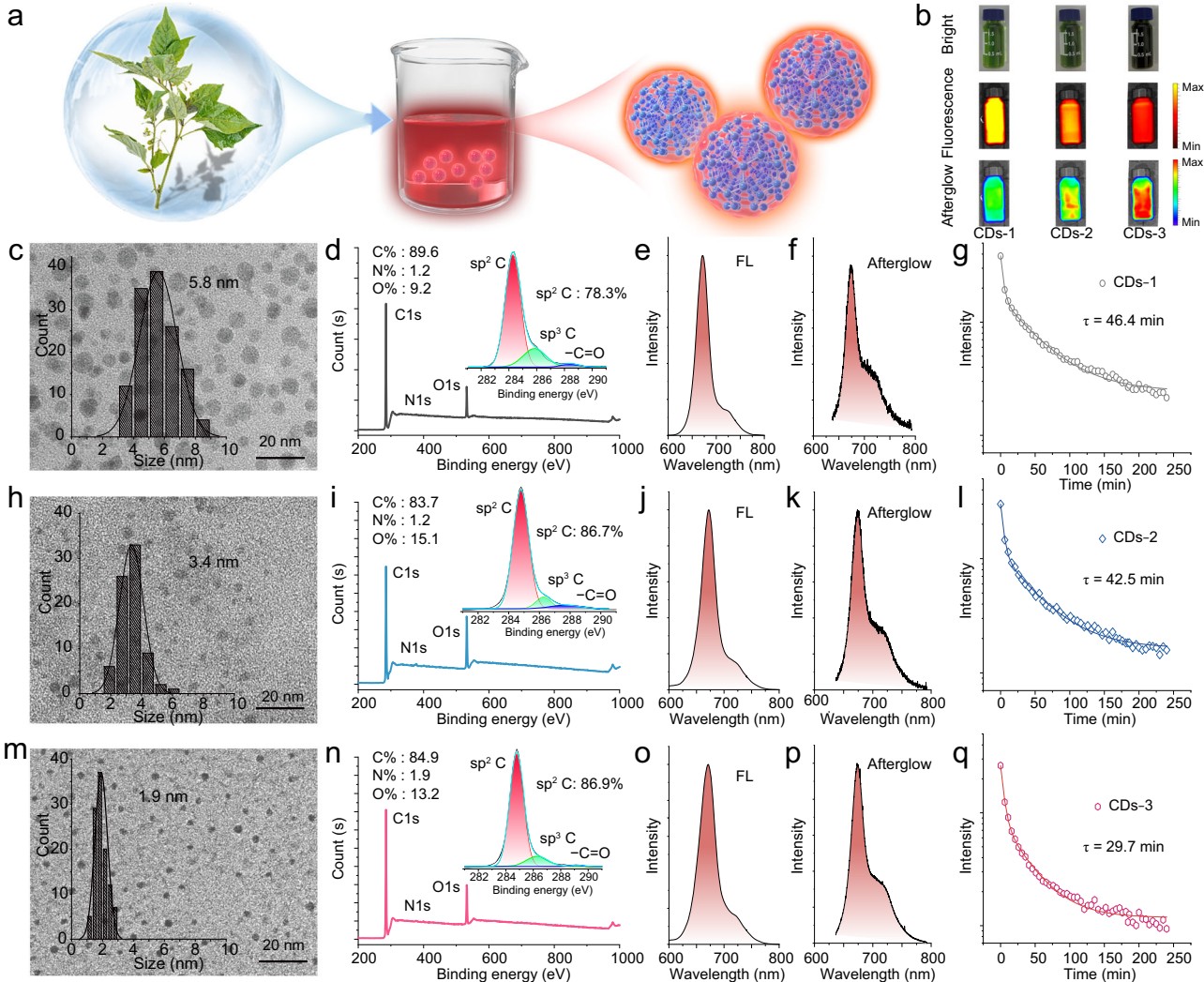

**Fig. 2 | Synthesis of the CDs and their luminescence. a** Schematic illustration of the preparation process of CDs. **b** Natural-light (up panel), fluorescence (middle panel) and afterglow luminescence (bottom panel) images of the CDs-1, CDs-2 and CDs-3. **c** Transmission electron microscopy images of CDs-1 (scale bars: 20 nm). **d** Full survey X-ray photoelectron spectroscopy (XPS) and high-resolution XPS C1s spectrum (inset) of the CDs-1. **e** Fluorescence spectra of CDs-1. **f** Afterglow spectrum of the CDs-1. **g** Afterglow lifetime curves of the CDs-1. **h** Transmission electron microscopy images of CDs-2 (scale bars: 20 nm). **i** Full survey X-ray photoelectron spectroscopy (XPS) and high-resolution XPS C1s spectrum (inset) of the CDs-2. **j** Fluorescence spectra of CDs-1. **k** Afterglow spectrum of the CDs-2. **l** Afterglow lifetime curves of the CDs-2. **m** Transmission electron microscopy images of CDs-3 (scale bars: 20 nm). **n** Full survey X-ray photoelectron spectroscopy (XPS) and high-resolution XPS C1s spectrum (inset) of the CDs-3. **o** Fluorescence spectra of CDs-3. **p** Afterglow spectrum of the CDs-3. **q** Afterglow lifetime curves of the CDs-3. Note: the condition of light irradiation is kept with 660 nm, 1.5 W cm$^{-2}$, 2 min, and the concentration of CDs: 1 mg mL$^{-1}$. The imaging in **c**, **h**, and **m** derived from three independent measurements, and the statistical distribution in **c**, **h**, and **m** derived from 100 independent measurements.

after light excitation, in which a series of oxidation reactions occur on their surface to induce the rupture and restructuration like C=C bonds of the CDs. Thus, chemical energy may release during the process, rendering CDs as 'energy storage batteries' to cause the long-time afterglow emission.

To further ascertain the mechanism of afterglow emission process, the fluorescence, absorption spectra and femtosecond transient absorption (TA) spectra of the CDs before and after light irradiation have been detailly characterized. As shown in Fig. 3g, h, the fluorescence intensity of CDs trends to decrease with the exposure of light irradiation as well as an attenuation in the absorption peak of CDs at 660 nm, declaring that the photophysical properties of CDs also vary with the light irradiation. Furthermore, TA spectroscopy of CDs before and after light irradiation was carried out under 410 nm light excitation. Supplementary Fig. 26 depicts TA spectra with probe wavelength at from 440 to 700 nm and delay time from 0.1 ps to 7 ns of CDs without light irradiation. The strong negative features from 645 to

700 nm correspond to ground state bleaching (GSB) and stimulated emission (SE) with peaks centered at 670 nm, which is consistent with the steady-state fluorescence and absorption spectra. The weak positive (red) features from 440 to 640 nm correspond to the excited state absorption (ESA). The TA spectra with different time delays have been illustrated in Fig. 3i. The negative peaks of SE centered at 670 gradually increase within 100 ps. The kinetic traces at different wavelengths as a function of delay time are also presented in Fig. 3j. Global fitting results demonstrate two principal decay associated difference spectra (DADS) of CDs without light irradiation (Fig. 3k). The fitted lifetimes of 14.7 ps (DADS1, black curve) and 2.2 ns (DADS2, red curve) are ascribed to two relaxation channels of photon-generated carriers in CDs. Here, the 14.7 ps lifetime represents the decay of an internal conversion (IC) pathway. While, the 2.19 ns one is SE pathway, which is consistent with the time resolved decay spectra of the CDs (Supplementary Fig. 12)[42,45]. After the light irradiation, the TA spectrum of CDs has changed prominently (Supplementary Fig. 27). Specifically, the GSB and SE signals

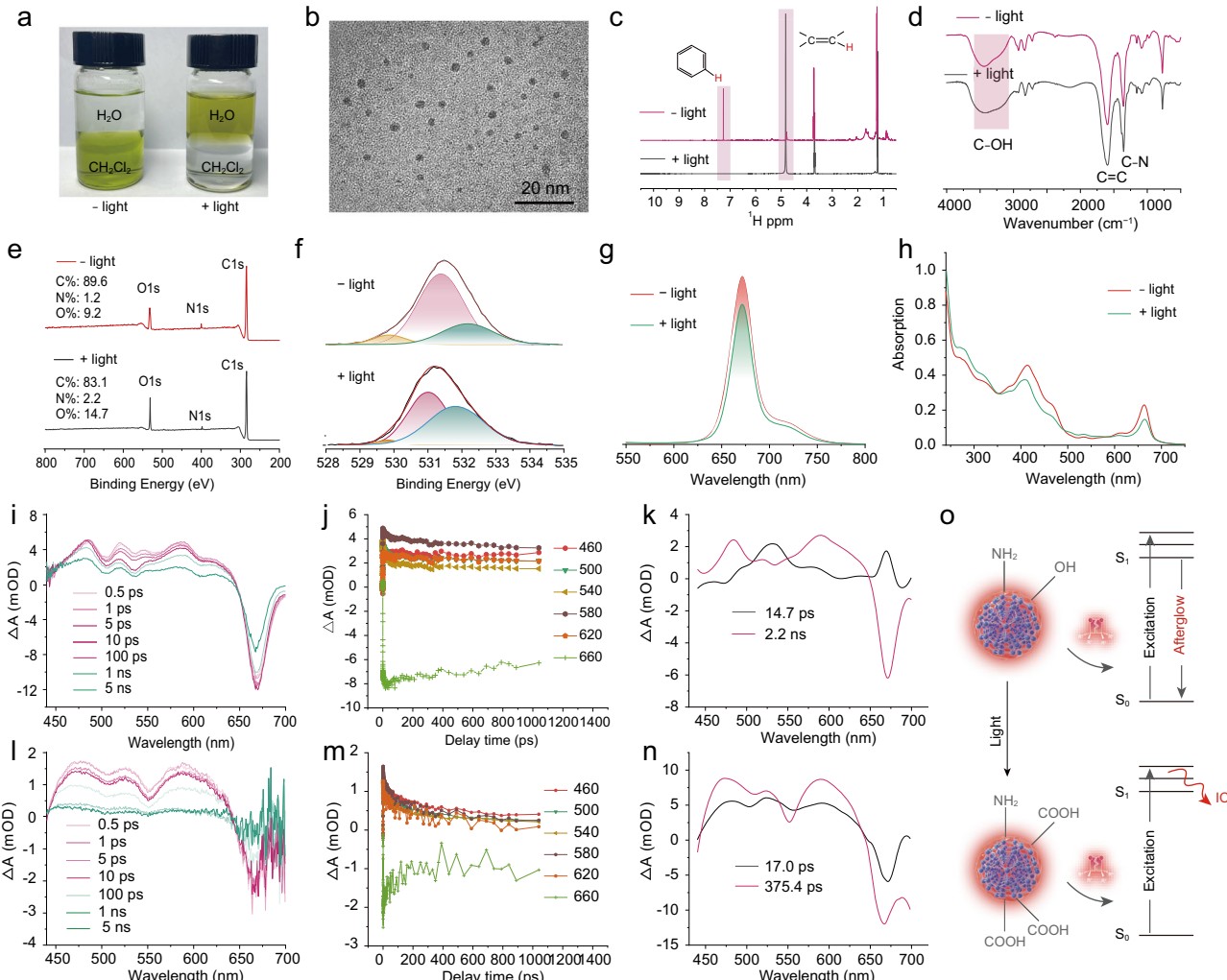

**Fig. 3 | Study on the structural and optical properties of CDs before and after light irradiation. a** The optical photograph of CDs in the mixed solution of water and dichloromethane before and after light irradiation. **b** Transmission electron microscopy images of CDs after light irradiation (scale bars: 20 nm). **c** $^1$H NMR spectra of the CDs before and after light irradiation. **d** Fourier transform infrared spectrum of the CDs before and after light irradiation. **e** Full survey XPS spectrum of the CDs before and after light irradiation. **f** The high-resolution XPS O1s spectrum of the CDs before and after light irradiation. **g** The fluorescence spectra of the CDs before and after light irradiation. **h** UV-vis absorption spectra of the CDs

before and after light irradiation. **i** Femtosecond transient absorption (TA) spectra of CDs at indicated delay times from 0.5 ps to 5 ns. **j** Kinetic traces at different probe wavelength of the CDs. **k** Global fitting TA results of CDs without light irradiation. **l** TA spectra of CDs after light irradiation at indicated delay times from 0.5 ps to 5 ns. **m** Kinetic traces at different probe wavelength of the CDs after light irradiation. **n** Global fitting TA results of CDs after light irradiation. **o** Schematic illustration of the change in luminescence of CDs after irradiation. Note: the condition of light irradiation is kept with 660 nm, 1.5 W cm$^{-2}$, 2 min. The imaging in **b** derived from three independent measurements.

as well as the kinetic trace at 660 nm of the CDs are obviously weakened obviously compared with those without light irradiation (Fig. 3l, m). And the global fitting curves present two fast delay channels with the lifetimes of 17.0 and 375.4 ps, respectively (Fig. 3n). The channel of 17.0 ps is consistent with the decay of the IC pathway of 14.7 ps lifetime before light irradiation. Whereas, the slow decay channel of 2.2 ns corresponding to radiative-transition process has disappeared and is replaced by a faster decay channel of 375.4 ps after light irradiation, manifesting that the photon-generated carriers from the CDs are prone to experience non-radiative transition via surface vibration relaxation[46]. On basis of the changes in the structure and photophysical properties of CDs before and after light irradiation, it can be concluded that the surface of the CDs has underwent a photo-induced oxidation process as illustrated in Fig. 3o. Concretely, hydrophobic functional groups such as phenyl groups on the surface of the CDs are oxidized to form carboxyl groups after light irradiation, causing the CDs to produce a lipophilic-hydrophily transition. In the meantime, light-emitting behaviors also vary largely with a

luminescence quenching due to the growing probability in nonradiative annihilation of electron hole pairs via more IC relaxation. Hence, the appearance of the photooxidation under illumination has a great impact on the light emission behavior of the CDs.

To further gain deep understanding of ultralong-persistent emission of the CDs, their luminescent behaviors and free radicals changes on the surface of the CDs under different light-irradiation time or temperature intervals have been performed. It can be seen from the afterglow emission curves that the afterglow emission intensity of CDs increases with the prolonging the exposure time of light irradiation (Supplementary Fig. 28). And even strong afterglow emission can maintain for the CDs after ceasing the photoexcitation for 2 h. In addition, the initial intensity of afterglow emission of CDs shows a trend of enhancement accompanied by a rapid attenuation in the total afterglow emission time of the CDs with raising the temperature (Supplementary Fig. 29). Then, the afterglow of the CDs by repeated light irradiation has been recorded. As depicted in Supplementary Fig. 30, the afterglow intensity of the CDs demonstrates an obvious

descending trend with four times of light irradiation (30 min each time), and only 32.4% of the initial intensity can be kept in the afterglow intensity with light-irradiation treatment. Therefore, the afterglow emission of the CDs is unrecoverable, whose irreproducibility is consistent with the structural and photophysical alteration of the CDs after light irradiation. In order to explore the reasons for the photo-induced superlong afterglow of the CDs, whose generating ability of $^1O_2$ free radicals at varied light-irradiation time and temperature has been further inquired. As illustrated in Supplementary Fig. 31, the CDs show obvious $^1O_2$ signals after light irradiation condition. With adding the light-irradiation time, the signal of $^1O_2$ presents an obvious enhancement trend (Supplementary Fig. 32). And the CDs at higher temperature also show the enhanced $^1O_2$ signal under the same light-irradiation time (Supplementary Fig. 33). Thus, the light irradiation- and temperature-dependent $^1O_2$ generation capacity has a positive correlation with the afterglow emission behavior of the CDs under similar conditions. In addition, the afterglow emission of solid-state CDs also confirms that its afterglow is related to oxygen rather than triplet exciton-related phosphorescence emission. Compared the solid-state CDs demonstrating obvious afterglow emission in air, the CDs encapsulated with epoxy resin show almost no afterglow after light irradiation (Supplementary Note 7, Supplementary Figs. 34 and 35). This can clearly verify that the afterglow of CDs requires the participation of oxygen. Different from triplet exciton-related phosphorescence emission, this photooxidation afterglow originates from the continuous oxidation of CDs by $^1O_2$ or other ROS species (Supplementary Note 8 and Supplementary Fig. 36), and oxygen is the active involvement in this oxidation process. Hence, the photooxidation approach can be used to elucidate the afterglow emission of the under light illumination and serve as a way to achieve ultralong-persistent afterglow.

### Theoretical calculation verification of the photooxidation triggered afterglow emission mechanism

Density functional theory (DFT) calculations have been also performed to earn insights into the underlying mechanism of the photooxidation ultralong afterglow with size-dependent afterglow lifetime. As shown in Fig. 4a, three simplified calculation units (IM0: C3, C5 and GNR) to simulate different CD sizes have been employed for DFT calculations, the process and atomic coordinates of the models are provided in Supplementary Note 9 and Supplementary Data 1. Calculated results verify that the introduction of $^1O_2$ facilitates $\pi^2$-$\pi^2$ cycloaddition whereby the oxygen molecule attacks the C=C bond, thus, leading to the production of dioxetane intermediate (IM1), which is unstable and facilely undergoes O−O bond cleavage to form the IM2 product. Furthermore, with the aid of $^1O_2$, the final product (IM3) can be generated with the benzene ring further oxidized to carboxyl[47–49]. The overall reaction process is exothermic with the energy of 8.3–9.1 eV, which is enough to excite the afterglow emission of the CDs. Based on the above results, we propose a probable photooxidation steps governing afterglow emission process of the CDs as illustrated in the Fig. 4b. In particular, (1) under light irradiation, ground-state electrons of the CDs can transit to excited state and the excited electrons can leap to ground state via the radiative recombination or the non-radiation transition, like the electron exchange with dissolved oxygen to form $^1O_2$. (2) Due to the appropriate oxidation of $^1O_2$, the C=C in CDs can be oxidized via $\pi^2$-$\pi^2$ cycloaddition to form a high-energetic dioxetane intermediate. (3) This intermediate is can further decompose by spontaneous oxidation to release a lot of chemical energy. (4) The chemical energy can transfer from intermediate to the CDs through electron exchange and further promote the ground-state electrons of the CDs to excited state. (5) The excited CDs then transit to the ground state and produce photon via a radiative recombination. With this continuous photooxidation process, the CDs can produce the long-persistent afterglow. Unlike common singlet/triplet states, charge-

separated states or activable trap states involved in regular traditional organic/inorganic afterglow materials, the intermediates produced by photooxidation may process a longer lifetime to induce the ultralong-persistent afterglow emission. Interestingly, compared to small-sized C3 units, larger-sized C5 and GNR units, especially GNR, have higher activation energy in the oxidation process of C=C due to the steric hindrance effect of quantum dots[50–52]. Thus, as the sample size increases, the energy barrier for oxidation is higher and the oxidation reaction is more difficult to proceed, which means a slower intermediate production rate to further result in a longer afterglow lifetime. Thus, although the initial afterglow intensity of CDs-1 with large size is lower than that of CDs-2 and CDs-3, the afterglow decay rate of CDs-1 is significantly slower than that of CDs-2 and CDs-3 owing to the steric hindrance effect (Supplementary Note 10 and Supplementary Fig. 37).

### Hydrophilic modification for the afterglow emissive CDs

The special photooxidation induced luminescence mechanism enables the CDs to effectively avoid dissolved oxygen induced quenching, thereby achieving ultralong-persistent NIR afterglow in aqueous solutions. In order to further improve water solubility and facilitate biological applications, a coating modification with amphiphilic Pluronic F127 has been performed here on the surface of the CDs. Here, we still use CDs-1 as an example. As shown in Fig. 5a, the CDs-1 is coated with F127 to form the polymer-crosslinked-CDs (p-CDs). The TEM and HR-TEM images present that the CDs are fully covered by F127 with a monodispersed size distribution and an average diameter of about 87.8 nm (Fig. 5b), and there are no Aggregation Induced Emission events in the p-CDs (Supplementary Note 11 and Supplementary Figs. 38–41. And the well-resolved lattice spacing of 0.21 nm, being ascribed to the (100) graphite crystallographic plane, can also be seen in the p-CDs. As determined by dynamic light scattering (DLS) in Fig. 5c, the dimensions of p-CDs-1, p-CDs-2, and p-CDs-3 are essentially identical at 223.1, 240.6, and 208.6 nm, which is fit for biotargeted imaging and diagnostic applications. As shown in Supplementary Fig. 42, there is nearly no significant difference in EPR intensity for the three samples, indicating the similar capability of p-CDs-1, p-CDs-2, and p-CDs-3 to generate $^1O_2$ under the same light irradiation (Supplementary Note 12). As illustrated in Fig. 5d, the afterglow emission intensity increases with the incremental concentrations of the p-CDs, presenting a significant linear growth relationship. And Fig. 5e has proved the changes in afterglow emission intensity of p-CD after light irradiation with different time intervals. As the light-irradiation time increases, the afterglow intensity of p-CDs has a significant enhancement, where the luminescent intensity with 5 min light irradiation is almost 2.2 times higher than that with 1 min light irradiation. In addition, due to the oxidation rate of the CDs decreases after coating, the afterglow intensity of p-CDs is obviously weaker than the initial CDs (Supplementary Note 13 and Supplementary Fig. 43). The afterglow lifetime of p-CDs is also tested. Intriguingly, size-dependent afterglow lifetime evolution from 3.4 to 5.9 h has been observed from the CD systems in aqueous solution (Fig. 5f, and Supplementary Table 1), which is the highest value in most of the organic, inorganic or the CD-based afterglow emission systems reported so far (Fig. 5g and Supplementary Table 2). Compared to the CDs without hydrophilic modification, p-CDs have an elongated afterglow lifetime, which can be attributed to that the coating of F127 may slow the production of reactive oxygen species, reduce the oxidation rate of the CDs, and finally increase the duration of light emission time via a retardant energy transfer from chemical energy to luminous energy. In addition, the afterglow emission stability of the p-CDs has been also investigated. Supplementary Figs. 44 and 45 show the afterglow intensity of the p-CDs is basically unchanged in acidic environment, and there is no obvious afterglow quenching phenomenon in common metal ion environment, hinting that the p-CDs have excellent afterglow emission stability. Moreover, the tissue penetration depth of the afterglow of

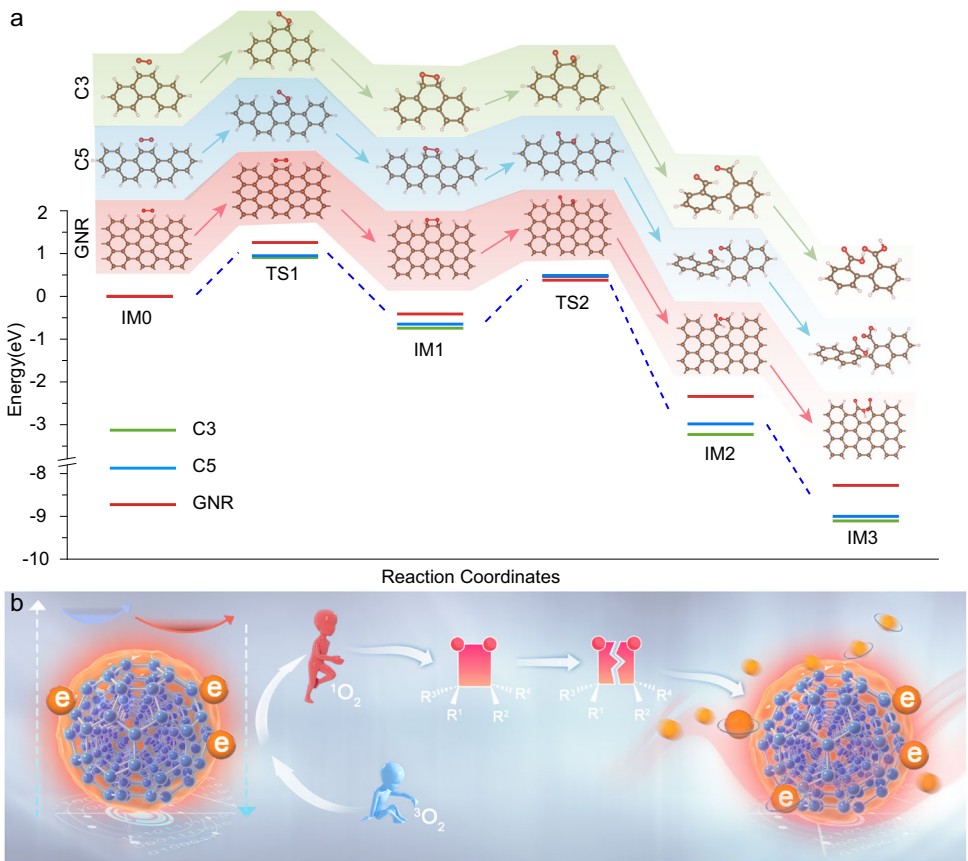

**Fig. 4 | Mechanism for the ultralong afterglow emission from CDs. a** DFT calculation for the reaction pathways of the CDs and $^1O_2$. **b** Possible mechanism proposed for photooxidation triggered afterglow of the CDs.

p-CDs has been also appraised, where the afterglow emission signal demonstrates a declining trend with increasing the thickness of chicken tissue (Fig. 5h). And the afterglow signal can be still seen at the tissue depth of 20 mm, which is beneficial for the bioimaging applications.

### In vivo biosafety and tumor targeting analysis of p-CDs

To facilitate biomedical application, the toxicity of the p-CDs in vivo was systematically evaluated to ensure their biosafety. Firstly, we have estimated their toxicological effects at the cellular level. As shown in Fig. 6a, nearly 100% viability for EC-9706 cells can be obtained when the concentration of the p-CDs is up to 1 mg mL$^{-1}$, indicating the high biocompatibility of the p-CDs, which also confirmed in different tissue cells (Supplementary Note 14 and Supplementary Fig. 46). Secondly, the p-CDs have been administered orally to mice for 14 days at 0.5 mg mL$^{-1}$ as a therapeutic dose to conduct the safety evaluations at the level of living animals. As illustrated in Fig. 6b, the weight change trend of the mice fed with the p-CDs aqueous solution is consistent with that of the control groups fed with equivalent pure water solution, and even the weight growth of mice fed with the p-CDs aqueous solution exceeds that of the control groups after cultivation for 7 days. After feeding the p-CDs for 14 days, blood routine tests in mice were also used to analyze the biosecurity of their p-CDs (Fig. 6c), wherein no significant difference is observed between the p-CDs treated groups and the control groups in the levels of albumin (ALB), alkaline phosphatase (ALP), alanine transaminase (ALT), aspartate transaminase (AST), γ-Glutamyl Transferase (GGT), and urea, indicating that p-CDs exhibit minimal effects on the kidney and liver of the mice. Furthermore, for the mice fed with p-CDs for 14 days, the hematoxylin and eosin (H&E) staining major organs, including heart, liver, spleen, lung, kidney, and brain, reveals no signs of damage or notable injuries,

demonstrating the low tissue toxicity of the p-CDs (Fig. 6d). In addition, after injecting p-CDs solution into the tail vein of mice for 2 h, the urine of mice was collected and tested. As shown in the Supplementary Fig. 47, obvious fluorescence and afterglow signal can be observed in the urine, indicating that the p-CDs can be excreted normally from the living body after injection. Meanwhile, the mice were dissected after 24 h, and it can be observed that there are only trace residues in the lungs and pancreas except for the strong enrichment of p-CDs in the tumor (Supplementary Fig. 48). Hence, the CDs can be eliminated from the body through bodily fluids. These data confirm a good safety profile of the p-CDs at the therapeutic dose in healthy mice.

Furthermore, the precise tumor targeting of the p-CDs was investigated. As shown in Fig. 6e, the afterglow images show that p-CDs began to accumulate in the tumor after 1 h intravenous injection, and obvious afterglow signals could still be observed 24 h later, indicating highly selective tumor targeting ability of the p-CDs due to the enhanced permeability/retention effect. In contrast, the fluorescence of p-CDs could not be used to accurately reveal the tumor area due to the interference of tissue autofluorescence. This result demonstrates the potential of afterglow luminescence imaging ability over fluorescence. In addition, confocal microscopic images of tumor tissue sections show an obvious deep-red luminescent signal of p-CDs, indicating that p-CDs can effectively enter and accumulate in tumor tissues (Fig. 6f). The good biosafety and precise targeting of the p-CDs demonstrated the great advantages in in vivo imaging.

### In vivo afterglow imaging and guidance for surgical resection of tumors

The excellent ultralong afterglow emission, good biosafety, and favorable tumor targeting endow the p-CDs make with great potential in high-contrast bioimaging and instruction of tumor resection surgery

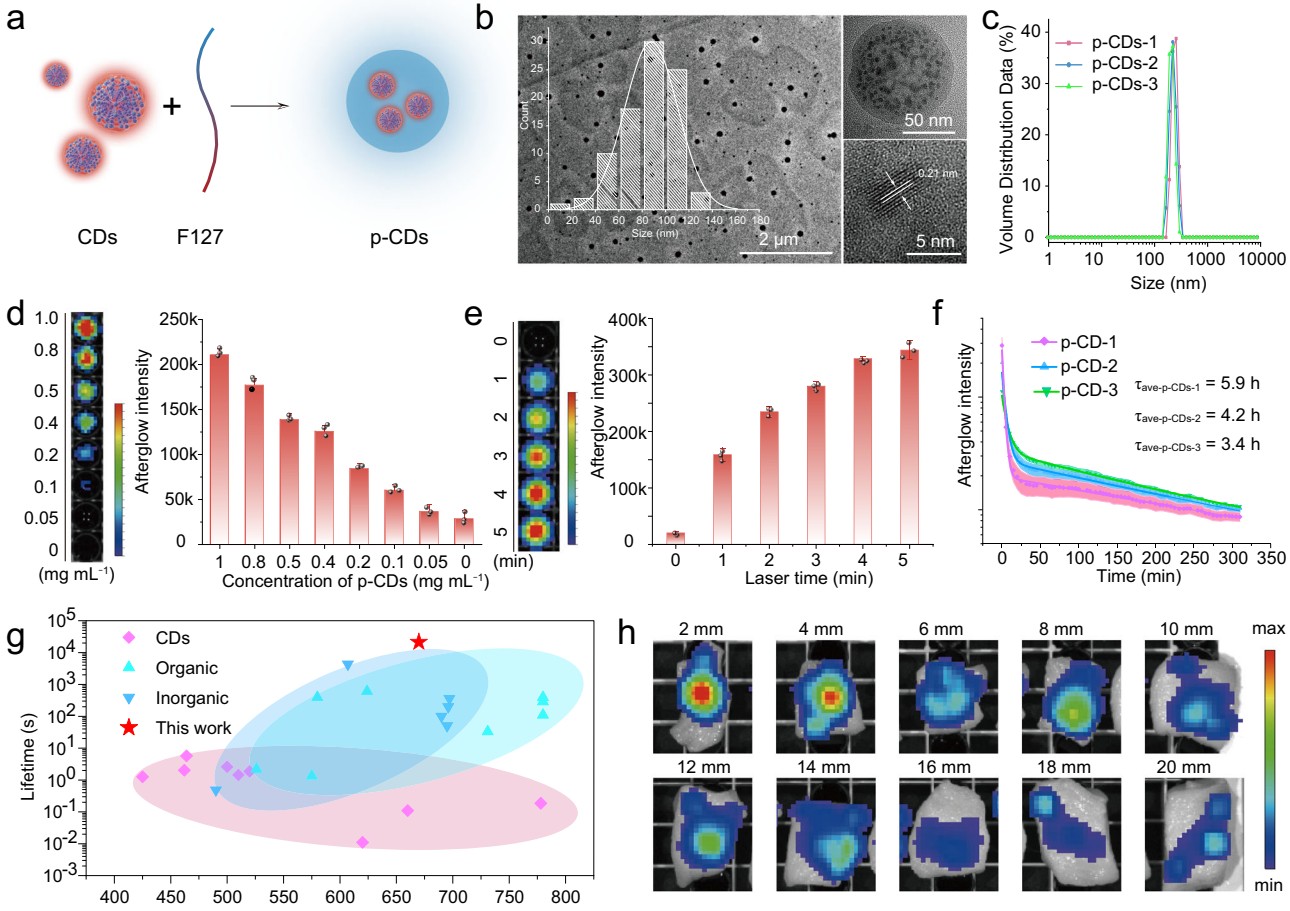

**Fig. 5 | Afterglow properties of the p-CDs. a** Schematic illustration of the preparation of the p-CDs. **b** Transmission electron microscopy (TEM, left, scale bars: 2 µm), high-resolution TEM (HR-TEM, upper right (scale bars: 50 nm) and lower right (scale bar: 5 nm)) images of the p-CDs. **c** Dynamic light scattering (DLS) distribution of the p-CDs in aqueous solution. **d** The afterglow photograph and the corresponding afterglow intensity of p-CDs with the concentration from 0 to 1 mg mL$^{-1}$ (660 nm, 1.5 W cm$^{-2}$, 2 min). **e** The afterglow photograph and the corresponding afterglow intensity of p-CDs with the time of light irradiation from 0 to 5 min (660 nm, 1.5 W cm$^{-2}$, 1 mg mL$^{-1}$). **f** Decay of afterglow of p-CDs over time at room temperature after light irradiation (660 nm, 1.5 W cm$^{-2}$, 1 mg mL$^{-1}$). **g** The lifetime of long afterglow materials (Supplementary Table 2). **h** Tissue penetration depths of the afterglow with a coverage of chicken breast issues with different thicknesses (the concentration of p-CDs: 1 mg mL$^{-1}$, 660 nm, 1.5 W cm$^{-2}$, 2 min). The imaging in **b** derived from three independent measurements, and the statistical distribution in **b** derived from 100 independent measurements. The error bars/bands presented in **d–f** indicate the statistical distribution derived from three independent measurements, the data show means + SD.

in vivo. As proof of principle, we have carried out experiments of afterglow imaging in vivo and guidance for surgical resection of tumors with the p-CDs as depicted in Fig. 7a. As shown in Fig. 7b, both the fluorescence and afterglow imaging for the mouse tumors after subcutaneous injection of the p-CDs have been acquired, where the fluorescence image of mice is greatly disturbed by the background light of its tissue, while the afterglow signal of tumor site is more obvious with high contrast. And the signal-to-noise ratio (SNR) through afterglow imaging for tumor is calculated to be 8.3, which is 3.6 times higher than that using the fluorescence signal for tumor imaging (Fig. 7c). These results verify further the potential of NIR afterglow of the p-CDs in vivo imaging. After subcutaneous injection of the p-CDs at the tumor site, the tumor proclaims obvious afterglow signal, and remains stable after continuous luminescence for 1 h (Fig. 7d, e). High SNR and long-persistent afterglow emission make it possible for the p-CDs to be used as imaging guidance in tumor resection. As shown in Fig. 7f, g, the afterglow imaging-guided process of the p-CDs in tumor surgery is proved with five steps. After the mouse tail vein injection of the p-CDs for 1 h, the position of the tumor can be clearly identified through the afterglow images after light irradiation, and the precise

operation on the tumor site of the mouse can be started with the imaging guidance. After the tumor site is exposed, the long-persistent afterglow of the p-CDs can still be used to accurately identify the location of the tumor as well as to achieve the complete removal of the mouse tumor via monitoring the afterglow emission signals from the mouse tissue (Fig. 7g). The above results have shown broad prospects for the NIR long-persistent afterglow p-CDs applied in biological imaging fields such as metastatic tumor localization, difficult surgical resection, etc.

## Discussion

With the high signal-to-background ratio, deep tissue penetration, and free of excitation/tissue self-fluorescence interference during the imaging process, NIR ultralong afterglow luminescence offers great advantages in biological imaging. Recent studies and reports have investigated the long afterglow emission of organic compounds and rare earth doped inorganic crystals. However, so far, there have been many problems in the afterglow emitters used in biological imaging, mainly focusing on the following issues: (1) The current materials often suffer from low biocompatibility. The introduction of rare earth and

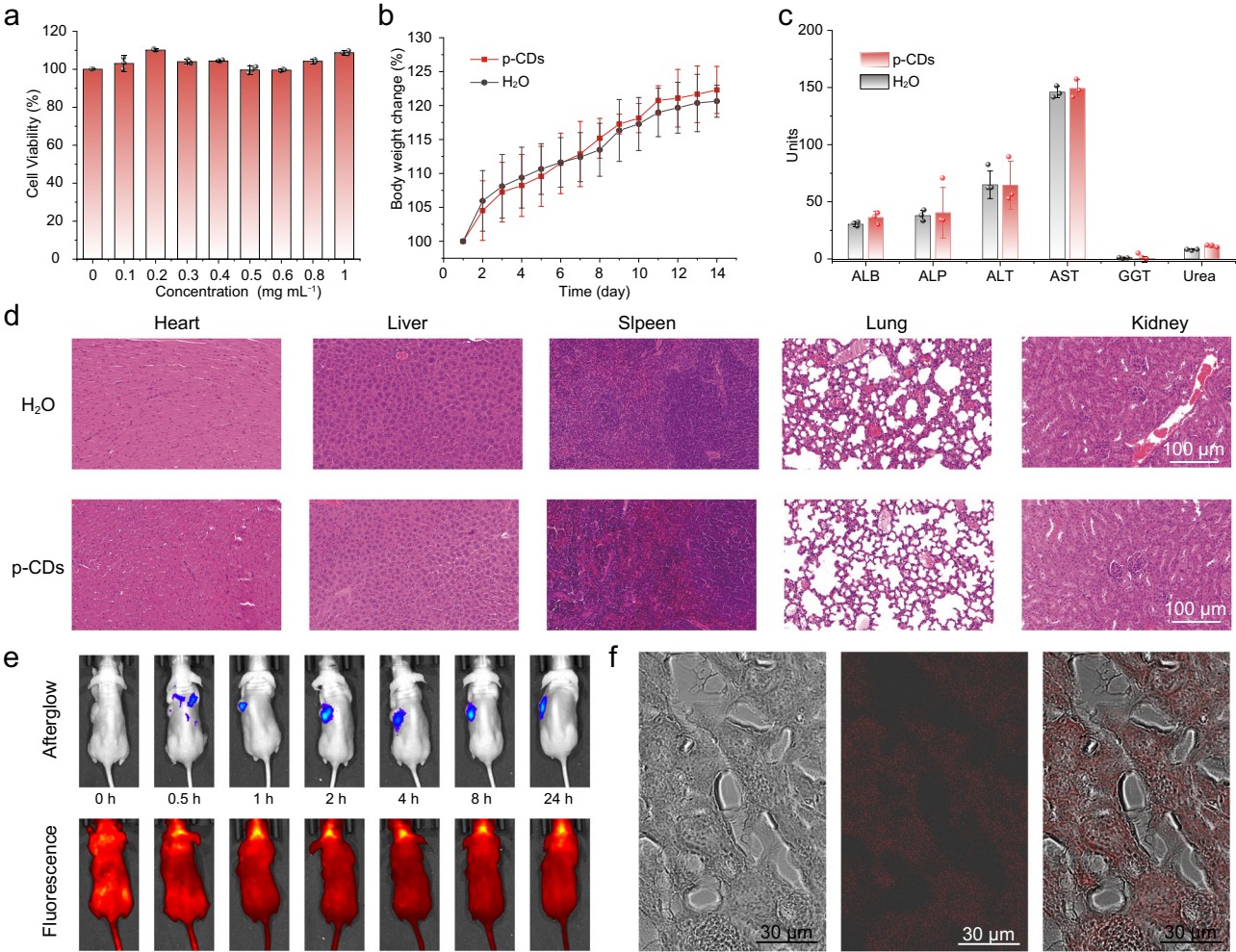

**Fig. 6 | In vivo biosafety and tumor targeting analysis of the p-CDs. a** Cell viability of EC-9706 cells after 24 h incubation in the different concentration of the p-CDs. **b** Body weight change in healthy mice after fed with $H_2O$ and p-CDs for 14 days, respectively ($n = 3$ mice). **c** Blood biochemical analysis test of healthy mice after fed with $H_2O$ and p-CDs for 14 days, respectively ($n = 3$ mice). **d** Hematoxylin-eosin staining images of the tissue sections of heart, liver, spleen, lung, and kidney, obtained from healthy mice after fed with $H_2O$ and the p-CDs for 14 days, respectively ($n = 3$ mice, scale bars: 100 µm). **e** Afterglow (upper) and fluorescence (lower) images of tumors of mouse after 24 h tail vein injection with the p-CDs. **f** The confocal microscopic images of tumor tissue sections (the red area corresponds to the fluorescence of p-CDs, scale bars: 30 µm). The error bars presented in **a**–**c** indicate the statistical distribution derived from three independent measurements, the data show means + SD. The imaging in **f** derived from three independent measurements.

precious metal ions brings significant biological toxicity, which prohibits its use in biological imaging. (2) The lifetime of afterglow luminescence is not long enough. The afterglow luminescence from organic molecules are mostly derived from the energy level transition between the singlet and triplet states, and the corresponding lifetime of this kind of emitters involved in the triplet-state excitons is usually limited in the range of microseconds to seconds. (3) The triplet excitons of organic materials can be easily quenched by dissolved oxygen in physiological water environments, hindering their practical applications in biological imaging. (4) In most cases, the afterglow luminescence with low influence from water is concentrated in the green light range, and its penetration depth is low.

Inspired by chemiluminescence generated by the electron transfer from the chemically induced high-energy intermediate in our previous reports[26,53–55], long-lifetime intermediates generated through photooxidation reaction have great potential in improving the afterglow duration of nanomaterials. And the photooxidation reaction can be effectively carried out in an oxygen containing environment, so as to prevent water quenching. As a kind of carbon-based luminescent nanomaterials, CDs have the characteristics of tunable wavelength,

high physicochemical inertness, low biological toxicity, and easy for preparation, manifesting great application prospects in the fields of biological imaging. CDs have sp²/sp³ hybrid conjugated structures and abundant heteroatom dopants, exhibiting excellent reactive oxygen generation and luminescence capabilities. In addition, the large adjustability in size and graphitization degree vests CDs with ultralong afterglow emission via steric hindrance effects, even overmatching traditional organic/inorganic afterglow materials. Therefore, utilizing the photooxidation afterglow mechanism can obtain NIR ultralong-persistent afterglow CDs.

In summary, we have achieved water-soluble NIR afterglow CDs with ultralong lifetime via a photooxidation strategy in this work. The CDs have obvious afterglow emission after irradiation with emission centers at 670 and 720 nm. DFT calculations also confirm that the size-dependent afterglow lifetime of the CDs can be attributed to the altered activation energy barrier via the steric hindrance effect. By further slowing down the light-induced-oxidation process via amphipathic molecular coating, an ultralong-lasting afterglow lifetime of 5.9 h can be achieved in the CDs, which is one of the highest values in currently reported organic/inorganic afterglow materials. With the

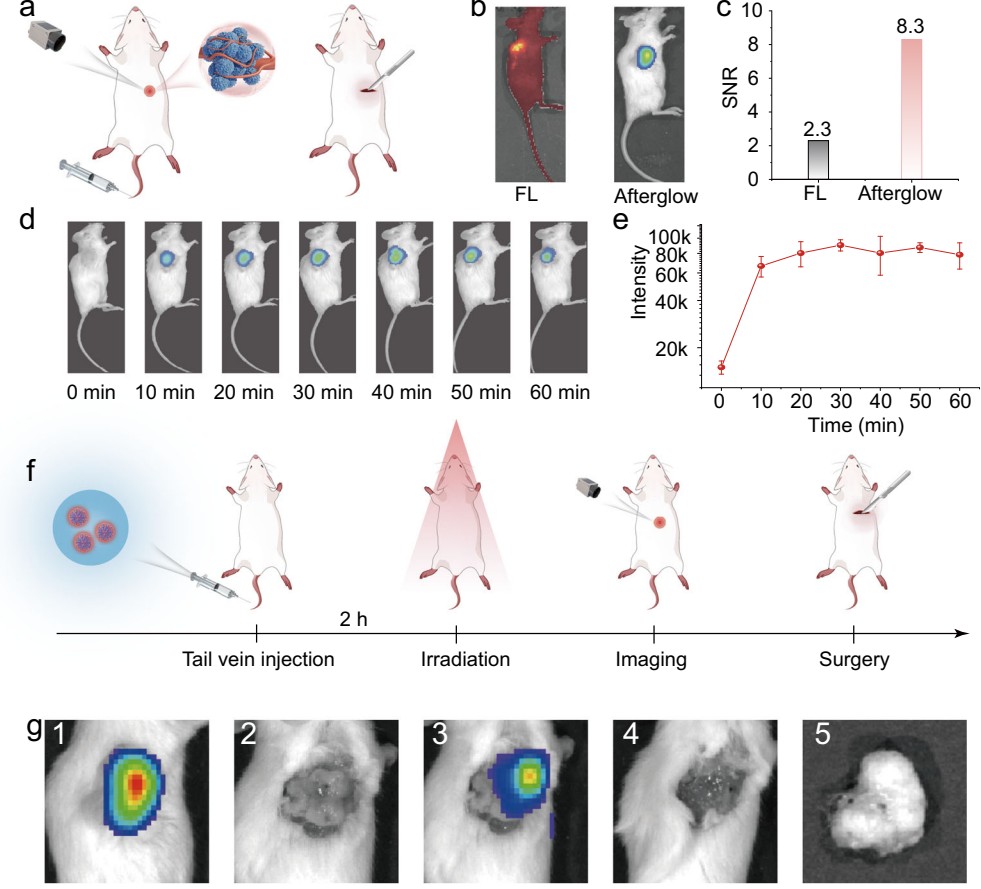

**Fig. 7 | In vivo afterglow imaging and guidance for surgical resection of tumors. a** The schematic illustrations of the p-CDs in vivo afterglow imaging and guidance for surgical resection of tumors. **b** The fluorescence and afterglow images of mouse tumor after subcutaneous injection of p-CDs (the concentration of p-CDs: 0.3 mg mL⁻¹, 100 μL, 660 nm, 1.5 W cm⁻², 2 min). **c** The corresponding signal-to-noise ratio (SNR) of the fluorescence and afterglow imaging in **b**. **d** The afterglow images of mouse at different time with the p-CDs subcutaneous injection (the concentration of p-CDs: 0.3 mg mL⁻¹, 100 μL, 660 nm, 1.5 W cm⁻², 2 min). **e** The corresponding intensity of afterglow images of mouse at different time in **d**. **f** The scheme of guidance for tumor surgical resection process. **g** The afterglow imaging monitoring and guiding the tumor surgery process. The error bars presented in **e** indicate the statistical distribution derived from three independent measurements, the data show means + SD.

photooxidative afterglow emission, dissolved oxygen quenching can be avoided, so CDs can also emit long-persistent luminescence in physiological environments for a long time, breaking through the obstacles in afterglow imaging related to rapid quenching. In addition, CDs exhibit great biological safety and deep tissue penetration depth of 20 mm, effective enrichment in the tumor site, and high image quality, which have been successfully used for long-time tumor-labeled imaging and precise surgical resection of mouse tumors via afterglow imaging guidance, proving the possibility of application in clinical settings.

## Methods
### Animal experiments
All animal procedures in this study were performed according to the National Institute of Health Guide for the Care and Use of Laboratory Animals. All animal studies were approved by the Experimental Animal Center of Henan Province (Zhengzhou, Hennan, China, SCXK-2017-0001), permit No. ZZU-LAC 20232331[12]. The maximal tumor size/burden permitted by our institutional review board is 15% of body weight (combined burden if more than one mass present) and mean tumor diameter = or >20 mm in adult mice (~25 g). The maximal tumor size/burden permitted by our institutional review board was not exceeded.

Twenty-four ICR mice (6–8 weeks) were taken by Henan Provincial Animal Experiment Center. Twenty-four Balb/c-nu/nu mice

(4–5 weeks) were provided by Beijing Vital River Laboratory Animal Technology. Animals were maintained in SPF conditions (24 ± 1 °C, 12 h light/dark cycle) with free access to water and chow.

### Materials
All chemicals and reagents were purchased from Macklin unless otherwise specified and were used without further purification. The *Solanum nigrum L* was purchased from Bozhou Kangyiyin Biotechnology Co., Ltd.

### Preparation of the CDs
The CDs were synthesized by the leaves of *Solanum nigrum L*, which was purchased from Bozhou Kangyiyin Biotechnology Co., Ltd., without undergoing any additional treatment. In detail, 1 g plant sources were dispersed into 20 mL ethanol (HPLC, ≥99.8%) and the mixture were transferred to a poly (tetrafluoroethylene) Teflon-lined autoclave (50 mL) and heated for 4 h at 80, 120 and 160 °C, respectively. After the products cooled down to room temperature, the solutions were filtered through a 0.22 μm polyether sulfone membrane to remove large particles, and the obtained crude products were further purified via a silica column chromatography with the eluent of ethyl acetate (HPLC, ≥99.8%) and petroleum ether ($V_{EA}$: $V_{PE}$ = 3:1) (Supplementary Note 15 and (Supplementary Fig. 49). After removing the solvent under reduced pressure, the CDs-1, CDs-2 and CDs-3 were collected for further characterization.

## Preparation of the p-CDs

To prepare p-CDs by an emulsion solvent evaporation method, 10 mg CDs was dissolved in 1.5 mL of dichloromethane or chloroform to form an oil phase, into which 9 mL of deionized water containing Pluronic127 (F127) (average Mn ~13,000) with 10 mg (i.e., a water phase) was slowly added. Emulsification was performed by probe sonication, with the sonication time of 10 min and a sonication power of approximately 100 W. After solvent evaporation of oil-in-water nanoemulsions by a rotary evaporator at room temperature, p-CDs were obtained by freeze-drying.

## Characterizations of CDs and p-CDs

The morphology and structure were characterized using an atomic force microscope (AFM, AIST-NT Smart SPM), and transmission electron microscopy (TEM, JEOL-2010). The crystalline property was evaluated in a Bruker-D8 Discover X-ray diffractometer with the Cu Kα line ($\lambda = 1.54\,\text{Å}$) as the irradiation source. The X-ray photoelectron spectroscopy (XPS) was measured on a Kratos AXIS HIS 165 spectrometer with a monochromatized Al KR X-ray source (1486.7 eV). Fourier transform infrared (FT-IR) spectroscopy was performed using a Thermo Scientific Nicolet iZ 10 spectrometer in the KBr tablets. Photograph was obtained from a D610 camera (Nikon, Japan). The fluorescence and afterglow spectrum of was measured by a F-7000 spectrofluorometer (Hitachi, Japan). The absorption spectrum was measured by an UV/vis spectrophotometer (Hitachi, UH-4150). The fluorescence decay curves were measured by Horiba FL-322 using a 410 nm Nano-LED monitoring the related emission. The Dynamic light scattering (DLS) distribution was measured with Zetasizer Nano ZSP in water. The PL QYs were measured by the spectrophotometer (FLS1000) in DMSO. The $^1H$ chemical shift was measured by nuclear magnetic resonance (NMR) (Bruker 400/600 M AVANCEIII) in CDCl$_3$. The photochemical free single oxygen was measured through electron spin resonance (EPR, Bruker A300). The specific types of irradiation sources are: ABI LED Light Bulb for Red Light Therapy, 660 nm Deep Red, 54 W Class, Amazon.

## The NMR spectra were acquired through the following procedure.

The chromatographic samples were incubated at 60 °C in a drying oven for 12 h. Subsequently, solvent removal was carried out, and some solid powder was dissolved in deuterated chloroform for NMR analysis. For the irradiated samples, separation was achieved using a mixture of methylene chloride and water for the preparation of NMR sample after the irradiation, and the obtained water-soluble CDs were freeze-dried to remove the aqueous fraction. The obtained solid-state irradiated samples were dissolved in D$_2$O (1 mg mL$^{-1}$) for NMR analysis.

MS spectrometry for the initial CDs and the CDs with 5 min irradiation was conducted by dissolving the samples in methanol (0.5 mg mL$^{-1}$) and subjecting them to MS analysis.

**EPR Test**. 500 mM of 2,2,6,6-Tetramethylpiperidine (TEMP) aqueous solution was prepared for EPR testing. Then, 2 mL of CDs (1 mg mL$^{-1}$, ethanol) were mixed with 0.5 mL of TEMP solution and the mixtures were conducted with EPR without light treatment. Additionally, the CD samples were irradiated under 660 nm, 2 W cm$^{-2}$ light for 2, 4 and 6 min and then conducted with EPR measurement. Furthermore, the original samples after light irradiation (660 nm, 1.5 W cm$^{-2}$, 2 min) were immersed in water bath at varying temperatures (20, 40, 60 and 80 °C) for 5 min before conducting EPR under identical conditions as above.

## The preparation of the sample of FT-IR.

The chromatographic samples were incubated at 60 °C in a drying oven for 12 h, followed by solvent removal and get the solid sample of CDs. The irradiated samples were then separated using a mixture of methylene chloride and water, with subsequent freeze-drying of the aqueous fraction into powder form.

## The FT-IR spectra acquired.

The prepared solid sample was evenly mixed with potassium bromide at a ratio of 1:100 and pressed, and then spectra acquired with a Fourier transform infrared spectrometer (Thermo Scientific Nicolet iZ 10). The air background is subtracted before the sample is tested.

## In vitro afterglow characterization of the CDs and p-CDs

100 μl of CDs and p-CDs with different concentrations (0, 0.05, 0.1, 0.2, 0.4, 0.5, 0.8 and 1.0 mg mL$^{-1}$) were placed in a black 96-well ELISA plate, with the different time (7, 6, 5, 4, 3, 2, 1 and 0 min) under certain light irradiation (660 nm, 1.5 W cm$^{-2}$). Then, the plate was immediately put into the in vivo imaging system (IVIS, Xenogen) to acquire afterglow luminescent signals with an exposure time of 2 min at every 5 min within 5 h with an open filter. Other in vitro afterglow tests were similar to those above.

## Cell culture

The human esophageal cancer cell line EC-9706, KYSE-150 and the human gastric cancer cells line MKN-45 were purchased from the Cell Bank of Type Culture Collection of Chinese Academy of Sciences (Shanghai, China). All the cells were cultured in Roswell Park Memorial Institute (RPMI) 1640 medium (Sigma, Sigma-Aldrich, St. Louis, USA, RNBJ4428) with 10% fetal bovine serum (FBS, Gemini, 900-108). The cells were maintained in an atmosphere of 5% CO$_2$ at 37 °C.

## In vivo safety evaluation

For establishment of in vivo safety evaluation models, ICR mice (6–8 weeks) were taken by Henan Provincial Animal Experiment Center. Before starting the experiment, animals were maintained in SPF conditions (24 ± 1 °C, 12 h light/dark cycle) with free access to water and chow. In the process of experiment, all mice were randomly assigned to one of two groups: test group or control groups. The mice in the test group were fed with p-CDs (0.1 mg mL$^{-1}$) whereas the mice in control group were fed with pure water. Body weight was measured every day following injection ($n = 3$ mice per group). Major organ tissues (heart, liver, kidney, spleen, and lung) and blood samples were collected at 14 days after fed with p-CDs and blood biochemical analysis was performed ($n = 3$ per group). Major organs were immediately fixed in 4% paraformaldehyde, then embedded in paraffin and sectioned into 5-μm slices. They were finally stained with hematoxylin-eosin.

## In vivo bioimaging with the p-CDs

For establishment of subcutaneous tumor models, Balb/c-nu/nu mice (4–5 weeks) were provided by Beijing Vital River Laboratory Animal Technology. Animals were maintained in SPF conditions (24 ± 1 °C, 12 h light/dark cycle) with free access to water and chow. After being acclimatized for 1 week, the A549 cells ($1 \times 10^7$) in 200 μL FBS free RPMI 1640 medium (50% Matrigel) were injected under the right forelimb armpit to the mice. Mice weight and tumor volume were monitored every 3 days, and the tumor volume was calculated as follows: ([short diameter]$^2$ × long diameter)/2. When the tumor grew to an average volume of 100 mm$^3$, and 100 μL p-CDs (0.3 mg mL$^{-1}$) was injected subcutaneously into the tumor of mouse. Then, the mouse was immediately put into the IVIS to acquire afterglow luminescent signals with an exposure time of 2 min at every 5 min within 1 h with an open filter. Similarly, mice were injected with 200 μL p-CDs (0.3 mg mL$^{-1}$) by tail vein injection, and the mouse was put into the IVIS to acquire afterglow luminescent signals with an exposure time of 2 min at 1 h and 24 h. The afterglow image of resected tumor was taken as above.

## Reporting summary

Further information on research design is available in the Nature Portfolio Reporting Summary linked to this article.

## Data availability

All data needed to evaluate the described conclusions are present in the manuscript and/or the Supplementary Information. The PL, absorption and afterglow spectra of CDs, afterglow lifetime of three kinds of CDs and p-CDs are provided in the source data. And the atomic coordinates of models (C3, C5 and GNR) are provided in the Supplementary Data 1. Data are available from the corresponding authors on request. Source data are provided with this paper.

## Code availability

All code used for analyzing the raw data is available upon request from the corresponding authors.

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

## Acknowledgements

We gratefully acknowledge the support of this research by the National Natural Science Foundation of China (12261141661, 12074348 and U2004168), and the Natural Science Foundation of Henan Province (212300410078).

## Author contributions

Q.L. and C.X.S. designed the project. G.S.Z., C.L.S. and C.Y.N. performed the experiments, calculation and wrote the manuscript. T.C.J., P.F.L., X.J.S. and Z.C. performed in vitro and in vivo experiments. R.W.S., Y.D., C.F.L., K.K.L., J.H.Z., and L.D. synthesized the carbon nanodots and carried out structural and optical characterization. All authors discussed the results throughout the project and approved the final version of the manuscript.

## Competing interests

The authors declare no competing interests.
