## [Peer Review File · Nature Communications]

Photooxidation triggered ultralong afterglow in carbon nanodotsREVIEWER COMMENTS

Reviewer #1 (Remarks to the Author):

The manuscript entitled “Photooxidation triggered ultralong afterglow in carbon nanodots” by Zheng and co-workers describes the synthesis and use of carbon dots retrieved from *Solanum nigrum* L by a solvothermal approach. The authors propose an original approach to the use of carbon dots for bioimaging based on their afterglow properties, rather than the most frequently studied fluorescence.

The study includes *ex vivo* and *in vivo* imaging of the carbon dots and provides an extensive morphochemical and optical characterization of the nanoparticles prior and after irradiation. The study is further completed with DFT calculations to support their hypothesis regarding the induced afterglow after photooxidation events occurring on the surface of the carbon dots.

The concept is intriguing and interesting but there are multiple aspects of the manuscript that should be clarified and/or addressed prior to further consideration for acceptance in *Nature Communications*:

1) The authors selected a precursor from biomass. Additional details regarding the preparation of the plant prior to the treatment should be clearly provided in detail. In addition, the authors should mention about potential variations of the properties of the carbon dots from batch to batch. Additional details about the source/supplier of the plant should be provided to ensure replicability elsewhere.

2) The main concept described in Scheme 1 and lines 122-127 should be more clearly described. The context of application and the role of oxygen in tumor environments where hypoxia can be a challenging issue is not clearly addressed.

3) Additional comments regarding production yields and quantum yields of the carbon dots should be included in the revised version of the manuscript.

4) The authors should clearly discuss how they discard the potential formation of 2 different types of carbon dots or the presence of organic fluorophores that can be potentially contributing to the double absorption detected in the samples.

5) The authors should also clarify and further discuss the nature of the N species and show their potential evolution before and after irradiation. These can be potential emitting centers that have not been taken into consideration. Additional experimental details (i.e. in the form of XPS) and additional discussion with the existing literature should be included in the revised version of the manuscript.

6) Please include a thorough description of the purification process: column supplier, column packing length, eluting volumes, collection times, fractions collected.

7) The authors should also clearly state when excitation of 410 nm or 660 nm is necessary, specifying the type of irradiation source (and manufacturers) used.

8) A more complete description of the NMR and MS spectra acquisition should be provided in the revised version. Additional experimental details regarding the EPR analysis should be further completed to ensure reproducibility elsewhere.

9) Have the authors considered Aggregation Induced Emission events in the p-CDs? Have the authors performed control measurements only with F-127? Can the authors provide the UV-Vis spectrum of the p-CDs?

10) Other minor questions:

i) Revise Figure 1, insets in XPS survey are not visible, the particle size distributions are also difficult to see.

ii) Expand the explanation to support Figure 3b

iii) Check spelling in L. 273, L.305, L. 447, L. 555

iv) Analysis by AFM and SEM is claimed but not shown.

v) Provide assignation of FT-IR bands in the graphs and expand explanation on how the sample is prepared and the spectra acquired.

Reviewer #2 (Remarks to the Author):

Ultralong afterglow emission in organic or inorganic materials usually involves the slow release of trapped charge in a crystal defect or a complicated protection approach for triple-state excitons. However, issues like potential toxicity, poor processability, and serious quenching from oxygen and water still hinder their practical application. The manuscript by Shan and coworkers describes a novel photooxidation strategy to construct extra-long afterglow emissions from carbon nanodots with a lifetime of up to an hour scale. This is a feat in the field of long persistent luminescence. Subsequently, afterglow imaging-guided surgery has been successfully performed to remove the tumor tissues accurately with carbon nanodots, demonstrating their clinical value. We believe this work is meaningful, and well-grounded and will be of indisputable interest to the readership of Nature Communications. Nevertheless, there are still some scientific issues to be clarified. A revision is required to improve this work for publication in this journal.

Our comments and questions are listed as follows:

1) CDs in ethanol solutions with different sizes obtain different afterglow lifetimes, while the afterglow lifetime of p-CDs is further improved, but the article does not give the size of p-CDs-1, p-CDs-2, and p-CDs-3, please give the sizes of p-CDs-1, p-CDs-2, and p-CDs-3 respectively, and discuss the reasons for obtaining longer afterglow lifetimes.

2) In this paper, the mechanism of photooxidative afterglow is reported, which can effectively avoid the quenching of oxygen, and whether the CDs can achieve afterglow in the solid state, which is very helpful for the discussion of the afterglow mechanism of the CDs, please test the afterglow of the CDs in the solid state and explain it.

- 3) The biosafety of the CDs was tested by feeding them to mice, but their cytotoxicity seems to be selected only for one kind of tumor cells, the toxicity test of the CDs in different tissue cells should be completed.
- 4) The manuscript explains the afterglow lifetime of CDs with different sizes as follows: “Interestingly, compared to small-sized C3 units, larger-sized C5 and GNR units, especially GNR, have higher activation energy in the oxidation process of C=C due to the steric hindrance effects of quantum dots. Thus, as the sample size increases, the energy barrier for oxidation is higher and the oxidation reaction is more difficult to proceed, which means a slower intermediate production rate to further result in a longer afterglow lifetime.” According to this theory, the initial afterglow of C3 should be stronger than that of C5 and GNR, please give the initial afterglow luminous intensity of the three to verify the correctness of this theory.
- 5) The author should provide more evidence on the fluorescence mechanism of CDs, especially its luminescence source at 720 nm.
- 6) In biological experiments, what is the biological metabolic cycle of the long afterglow CDs? And they can be eliminated from the body through bodily fluids, or not.
- 7) After hydrophilic treatment, the lifetime of CDs increased by six times. Why is there such a significant improvement in their lifetime? Has the luminescence intensity of CDs of the same size also changed before and after treatment?
- 8) The reason for selecting the leaves of *Solanum nigrum* L to prepare such carbon dots should be given. I am curious whether other biomass raw materials can achieve similar results.
- 9) Is the ultralong afterglow emission in the manuscript related to the triplet-state radiative recombination of CDs? The authors should provide other scientific evidence to support this argument.

Reviewer #3 (Remarks to the Author):

In this manuscript, the authors report an interesting photooxidation triggered ultralong afterglow, surprisingly, it is possible to achieve an afterglow lifetime of up to 5.9h in carbon nanodots. Conceivably, this approach can be used to the development of new-type afterglow materials without water quenching or toxicity problems. I think this is a solid and important piece of work in the field of long persistent phosphors, which will no doubt have a significant impact for the scientific community of Nature Communications. I recommend publication after addressing the following minor points:

1. In this work, the morphology of CDs-1, CDs-2, and CDs-3 are only investigated with TEM. The high-resolution TEM image and AFM should also be provided, which can further illustrate the exact morphologies of the as-prepared CDs.
2. The authors present “obvious afterglow with the wavelength extending to the NIR region can be observed from the CDs, whose afterglow spectra are consistent with their fluorescence spectra (Figs. 1f, 1k and 1p). However, the afterglow spectra of these CDs seem slightly different from the fluorescence. Can the authors provide the exact comparison and discuss the possible reason?
3. The authors present that “All the afterglow emission time of these CDs in ethanol solution is over 4 h, and the corresponding lifetimes are fitted to 46.4, 42.5 and 29.7 minutes (min) for CDs-1, CDs-2, and

CDs-3 (Figs. 1g, 1l and 1q)". However, the reproducibility of afterglow lifetime has not been provided. Can the authors provide the detailed experimental method and compared the results?

4. The author claim "photooxidation approach can be used to elucidate the unique afterglow emission". Whether the common oxidant can trigger the afterglow? This may be a more obvious evidences about the afterglow mechanism.

5. The authors present "photooxidation approach can be used to elucidate the unique afterglow emission" and "the energy barrier for oxidation is higher and the oxidation reaction is more difficult to proceed, which means a slower intermediate production rate to further result in a longer afterglow lifetime". However, the generation capability of $1O_2$ is also possible. The authors should provide more evidences to support the mechanism.

6. In Fig. 5a, nearly 100% viability for Hela cells can be obtained when the concentration of the p-CDs is up to $300 \mu\text{g mL}^{-2}$. However, when the concentration of the p-CDs reaches $400 \mu\text{g mL}^{-2}$, the cell viability seems to be lower than 80%. It is very confusing. The author can provide the IC_{50} to exactly confirm the high biocompatibility of the p-CDs.

7. The calculation results are quite interesting. But the description about the calculation is quite confusing. What's the exact code and software for theory calculation in S26? Can the author provide the computational detail, including structural optimization and calculation of transition state potential barriers?

Response to Reviewers' comments:

We are very grateful to the reviewers for their thoughtful, detailed and constructive comments on our manuscript (Manuscript ID: NCOMMS-23-42761). We have checked the comments carefully and revised the manuscript according to these comments. Revised portion is marked in red in the manuscript. The point-by-point response to the comments is highlighted in blue style and listed as follows:

Reviewer #1: The manuscript entitled “Photooxidation triggered ultralong afterglow in carbon nanodots” by Zheng and co-workers describes the synthesis and use of carbon dots retrieved from *Solanum nigrum* L by a solvothermal approach. The authors propose an original approach to the use of carbon dots for bioimaging based on their afterglow properties, rather than the most frequently studied fluorescence. The study includes ex vivo and in vivo imaging of the carbon dots and provides an extensive morpho chemical and optical characterization of the nanoparticles prior and after irradiation. The study is further completed with DFT calculations to support their hypothesis regarding the induced afterglow after photooxidation events occurring on the surface of the carbon dots. The concept is intriguing and interesting but there are multiple aspects of the manuscript that should be clarified and/or addressed prior to further consideration for acceptance in Nature Communications:

1) The authors selected a precursor from biomass. Additional details regarding the preparation of the plant prior to the treatment should be clearly provided in detail. In addition, the authors should mention about potential variations of the properties of the carbon dots from batch to batch. Additional details about the source/supplier of the plant should be provided to ensure replicability elsewhere.

Response: Thank you for the valuable suggestions. Firstly, the source/supplier of the plant (*Solanum nigrum* L) was purchased from Bozhou Kangyiyin Biotechnology Co., Ltd. (Fig. R1) to prepare the afterglow CDs without any additional treatment. In detail, 1 g plant sources were dispersed into 20 mL ethanol and the mixtures were transferred to a poly (tetrafluoroethylene) Teflon-lined autoclave (50 mL) and heated

for 4 h at 80, 120 and 160 °C, respectively. After the products cooled down to room temperature, the solutions were filtered through a 0.22 μm polyether sulfone membrane to remove large particles, and the obtained crude products were further purified via a silica column chromatography with the eluent of ethyl acetate and petroleum ether ($V_{EA} : V_{PE} = 3:1$). After removing the solvent under reduced pressure, the CDs-1, CDs-2 and CDs-3 were collected for further characterization.

Secondly, we have prepared three batches to investigate the potential variations of the properties of the CDs. As illustrated in Fig. R2 and R3, these batches of CDs have been checked with fluorescence and absorption spectra. And it can be observed that these three CDs exhibit almost same fluorescence and UV-vis absorption spectra without peak shift. Meanwhile, the PL QYs of each batch have been also compared, and the results reveal that three batches of CDs possess similar PL QYs of 27.69, 27.92 and 28.73% (Fig. R4), indicating the excellent repeatability of the preparation approach to obtain CDs from batch to batch.

We have added the above contents and figures in the revised manuscript and supplementary information.

Fig. R1. The photograph of *Solanum nigrum* L purchased from Bozhou Kangyiyin Biotechnology Co., Ltd.

Fig. R2. Fluorescence spectra of the CDs-1 prepared from three batches.

Fig. R3. UV-vis spectra of the CDs-1 prepared from three batches.

Fig. R4. PL QYs of the three batches of CDs-1.

2) The main concept described in Scheme 1 and lines 122-127 should be more clearly described. The context of application and the role of oxygen in tumor environments where hypoxia can be a challenging issue is not clearly addressed.

Response: Thank you for this helpful comment. The main concept described in Scheme 1 and lines 122-127 is the innovations and corresponding mechanism of photooxidation afterglow from CDs. We have further clarified the concept about the Scheme 1 as follows:

“Under irradiation from high-energy photons, the ground-state electrons of CDs can transit to excited state and then the excited electrons can leap to ground state via the

radiative recombination or the non-radiation transition, like the electron exchange with dissolved oxygen to form singlet oxygen ($^1\text{O}_2$). Due to the appropriate oxidation of $^1\text{O}_2$, the CDs can be oxidized and simultaneously produce one kind of energetic intermediates. The metastable intermediates can further slowly decompose and transfer the chemical energy to the CDs, resulting in the excitation and ulteriorly long-persistent afterglow. With the photooxidation-assisted afterglow, the issue about the dissolved oxygen-caused quenching of triplet excitons in common phosphorescence or thermal activation delayed fluorescence can be effectively prevented”.

On the other hand, the context of application and the role of oxygen in tumor environments where hypoxia can be a challenging issue are also further discussed. The corresponding address is as follows:

“Owing to the aggressive proliferation of cancer cells and insufficient blood supply in tumors, the tumor environment features angiogenesis, maladjusted biosynthesis intermediates, acidosis, and hypoxia. Actually, hypoxia is an important character of malignant tumors, which depends on tumor angiogenesis and rapid growth of tumor cells. The newly formed blood vessels are different from normal blood vessels, and the poor oxygen supply capacity and slow blood flow will lead to the insufficient oxygen and nutrients transported in tumor cells, resulting in an imbalance in oxygen supply and consumption in tumor tissue (Nat. Rev. Urol. 7, 258-266 (2010)). In addition, due to the abnormal metabolism of tumor cells, the tumor environment extensively presents excess reactive oxygen species (ROS) of H_2O_2 , $\bullet\text{O}^{2-}$ and $\bullet\text{OH}$ (Nat. Rev. Cancer 22, 280 (2022)) and the superfluous energy transfer between ROS and triplet excitons will seriously limit the bioimaging assisted by phosphorescence or thermal activation delayed fluorescence. Therefore, it is still a huge challenging issue to achieve the precise diagnosis and therapy of tumor under such complex oxygen environment. A facile strategy to achieve ultralong-persistent and high-quality afterglow imaging in intricate biological environment is using CDs as emitter via storing the excited energy in long-lifetime intermediate states or defects, and then tardily emitting photons after the removal of light irradiation (Scheme 1). With unique carbon conjugated structure, CDs can act well as both electron donors and acceptors.

Under irradiation from high-energy photons, the ground-state electrons of CDs can transit to excited state and then the excited electrons can leap to ground state via the radiative recombination or the non-radiation transition, like the electron exchange with dissolved oxygen to form singlet oxygen ($^1\text{O}_2$). Due to the appropriate oxidation of $^1\text{O}_2$, the CDs can be oxidized and simultaneously produce one kind of energetic intermediates. The metastable intermediates can further slowly decompose and transfer the chemical energy to the CDs, resulting in the excitation and ulteriorly long-persistent afterglow. With the photooxidation-assisted afterglow, the issue about the dissolved oxygen-caused quenching of triplet excitons in common phosphorescence or thermal activation delayed fluorescence can be effectively prevented”

We have added the above discussion in the revised manuscript.

3) Additional comments regarding production yields and quantum yields of the carbon dots should be included in the revised version of the manuscript.

Response: Thank you for the valuable suggestion. The production yields and quantum yields of the CDs have been tested. As illustrated in Fig. R5, we have prepared the CDs-1 from 1.0006 g plant source and obtained 0.0135 g CDs, thus the production yields of CDs is about 1.35%.

Fig. R5. Mass of the source, initial flask and the flask with the CD products.

Meanwhile, as illustrated in Fig. R6, with the directly integrating sphere measurement, the PL QYs of CDs-1, CDs-2 and CDs-3 are calculated as 28.7%, 31.8% and 31.0%, respectively.

We have added the above contents and figures in the revised manuscript and supplementary information.

Fig. R6. PL spectra with and without sample, and the PL QYs of **a** CDs-1, **b** CDs-2 and **c** CDs-3.

4) The authors should clearly discuss how they discard the potential formation of 2 different types of carbon dots or the presence of organic fluorophores that can be potentially contributing to the double absorption detected in the samples.

Response: Thank you for this helpful comment. In the process of solvothermal preparation, different CDs and other molecular fluorophores may be produced. However, the column chromatography purification can efficiently separate the required CDs from the potential formation of 2 different types of CDs or the presence of organic fluorophores (Fig. R7). We believe the column chromatography is one valid technology to obtain pure CDs as reported in previous literatures (Nat.

Photonics 14, 171–176 (2020)).

The detail of the preparation of CDs and the column chromatography purification: Firstly, the source/supplier of the plant (*Solanum nigrum* L) was purchased from Bozhou Kangyiyin Biotechnology Co., Ltd. to prepare the afterglow CDs without any additional treatment. In detail, 1 g plant sources were dispersed into 20 mL ethanol and the mixture were transferred to a poly (tetrafluoroethylene) Teflon-lined autoclave (50 mL) and heated for 4 h at 80, 120 and 160 °C, respectively. After the products cooled down to room temperature, the solutions were filtered through a 0.22 μm polyether sulfone membrane to remove large particles, and the obtained crude products were further purified via a silica column chromatography with the eluent of ethyl acetate and petroleum ether ($V_{EA} : V_{PE} = 3:1$). After removing the solvent under reduced pressure, the CDs-1, CDs-2 and CDs-3 were collected for further characterization.

Fig. R7. The photograph of the silica gel plate with the CDs in the eluent for the column chromatography.

5) The authors should also clarify and further discuss the nature of the N species and show their potential evolution before and after irradiation. These can be potential emitting centers that have not been taken into consideration. Additional experimental details (i.e. in the form of XPS) and additional discussion with the existing literature should be included in the revised version of the manuscript.

Response: Thank you for the valuable comment. As reported in previous literatures

(Adv. Mater. 2020, 1906641. Small 2018, 14, 1703919. Adv. Sci. 2022, 2203622), the fluorescence emission of CDs is related to the doping of N atoms. Generally, the pyridine N and pyrrole N can contribute to the formation of a π -conjugated system with a pair of p-electrons in the CDs, providing a strong radiative recombination center. To investigate the potential evolution of the CDs under light irradiation, we have compared the N species of the CDs before and after lasting irradiation. As shown in the Fig. R8 and R9, the N 1s spectra exhibit three peaks assigned to pyridinic N, pyrrolic N, and graphitic N for the CDs before and after light irradiation. And the corresponding content and type of N species did not change significantly, indicating that photooxidation primarily occurred on the surface of CDs. As a result, the non-N-related non-radiative recombination centers like $-\text{COOH}$ can be generated (Anal. Chem. 89, 2017, 12520–12526. Chem. Commun. 53, 2017, 2122–2125. Nanoscale 5, 2013, 2655–2658), leading to the slight reduction in fluorescence and afterglow intensity of CDs after the photooxidation process.

We have added the above discussion and figures in the revised manuscript and supplementary information.

Fig. R8. The full survey XPS of the CDs-1 before and after light irradiation.

Fig. R9. a The high-resolution XPS N1s spectrum of the CDs before light irradiation. **b** The high-resolution XPS N1s spectrum of the CDs after light irradiation.

6) Please include a thorough description of the purification process: column supplier, column packing length, eluting volumes, collection times, fractions collected.

Response: Thank you for the valuable comment. In this work, the optimally selected column parameters are as follow: sand plate chromatographic column, $\phi 46$ mm; effective length, 457 mm; joint aperture, 2 mm, 24/40, C184464C, synthware; column chromatography silica gel size, 200-300 mesh, 500 g/bottle, T242030, synthware. The column filling length was about 320 mm, and the elution volume was about 600 mL. As illustrated in Fig. R10, the position of the CDs can be observed under 365 nm light, and the sample was collected as the CDs flowed downward. About 40 mL of the collected CD solution was obtained, approximately constituting 1/15 of the elution volume.

Fig. R10. a The photograph of the silica gel plate with the CDs in the eluent for the column chromatography. **b** The photograph of the purification process (The red emission is produced by irradiation at 365 nm).

7) The authors should also clear state when excitation of 410 nm or 660 nm is necessary, specifying the type of irradiation source (and manufacturers) used.

Response: Thank you for the valuable comment. Due to the high tissue penetration depth of red light, the light source irradiated by the sample in vitro and in vivo was 660 nm. The specific types of irradiation sources are: ABI LED Light Bulb for Red Light Therapy, 660 nm Deep Red, 54 W Class, Amazon.

8) A more complete description of the NMR and MS spectra acquisition should be provided in the revised version. Additional experimental details regarding the EPR analysis should be further completed to ensure reproducibility elsewhere.

Response: Thank you for the valuable comment. The NMR spectra were acquired as follows: The chromatographic samples were incubated at 60 °C in a drying oven for 12 h. Subsequently, solvent removal was carried out, and some solid powder was dissolved in deuterated chloroform for NMR analysis. For the irradiated samples, separation was achieved using a mixture of methylene chloride and water for the preparation of NMR sample after the irradiation, and the obtained water-soluble CDs were freeze-dried to remove the aqueous fraction. The obtained solid-state irradiated samples were dissolved in D₂O (1 mg mL⁻¹) for NMR analysis.

MS spectrometry for the initial CDs and the CDs with 5 min irradiation was conducted by dissolving the samples in methanol (0.5 mg mL⁻¹) and subjecting them to MS analysis.

The EPR analysis was acquired as follows: 500 mM of 2,2,6,6-Tetramethylpiperidine (TEMP) aqueous solution was prepared for EPR testing. Then, 2 mL of CDs (1 mg mL⁻¹, ethanol) were mixed with 0.5 mL of TEMP solution and the mixtures were conducted with EPR without light treatment. Additionally, the CD samples were

irradiated under 660 nm, 2 W cm⁻² light for 2, 4 and 6 min and then conducted with EPR measurement. Furthermore, the original samples after light irradiation (660 nm, 1.5 W cm⁻², 2 min) were immersed in water bath at varying temperatures (20, 40, 60 and 80 °C) for 5 min before conducting EPR under identical conditions as above.

We have added the above details in the revised supplementary information.

9) Have the authors considered Aggregation Induced Emission events in the p-CDs? Have the authors performed control measurements only with F127? Can the authors provide the UV-Vis spectrum of the p-CDs?

Response: Thank you for the valuable suggestion. The PL and UV-vis spectra of p-CDs and CDs have been tested. As shown in Fig. R11, the p-CDs present the same PL spectra as the initial CDs. Meanwhile, the PL intensity of p-CDs at various concentrations was compared. As illustrated in the Fig. R12, with an increase in p-CDs concentration, the PL intensity initially enhances and subsequently diminishes, verifying that there are no Aggregation Induced Emission events in the p-CDs. Meanwhile, the p-CDs prepared with only F127 illustrate nearly no fluorescence (Fig. R11a), verifying that the weak influence of F127 to prepare the afterglow p-CDs. In addition, the UV-Vis absorption spectrum of p-CDs has been also measured. As illustrated in Fig. R13, there are no significant difference between the CDs and p-CDs in their absorption spectra, indicating uniform dispersion of CDs in F127 rather than in interparticle aggregation state. This result is also supported by the TEM results from CDs and p-CDs (Fig. R14), in which the CDs are evenly and homogeneously dispersed in the F127 polymer.

Fig. R11. a Fluorescence spectrum of p-CDs and F127, **b** Fluorescence spectra of CDs.

Fig. R12. The FL intensity of p-CDs at various concentrations.

Fig. R13. UV-Vis spectra of **a** p-CDs, **b** CDs and **c** F127.

Fig. R14. Transmission electron microscopy (TEM, left), high-resolution TEM (HR-TEM, upper right) images and selected area electron diffraction (SAED, lower right) pattern of **a** CDs and **b** p-CDs.

10) Other minor questions:

- i) Revise Figure 1, insets in XPS survey are not visible, the particle size distributions are also difficult to see.
- ii) Expand the explanation to support Figure 3b.
- iii) Check spelling in L. 273, L.305, L. 447, L. 555.
- iv) Analysis by AFM and SEM is claimed but not shown.
- v) Provide assignation of FT-IR bands in the graphs and expand explanation on how the sample is prepared and the spectra acquired.

Response: We are very grateful for your careful reading and the detailed suggestions. We have carefully checked the figures, grammar, spelling, and the description of this manuscript. In the revised manuscript, all detectable mistakes have been corrected.

- i) We have corrected Fig. 1 and adjusted its resolution to make it clearer (Fig. R15). However, due to the submission system with a conversion process from Word to PDF, the image may be compressed and the pictures present quite low resolution. In the revised version, we have included the original picture separately and uploaded it.

Fig. R15. The revised Fig 1.

ii) Figure 3b is explained in detail as follows: (1) Under light irradiation, ground-state electrons of the CDs can transit to excited state and the excited electrons can leap to ground state via the radiative recombination or the non-radiation transition, like the electron exchange with dissolved oxygen to form singlet oxygen ($^1\text{O}_2$). (2) Due to the appropriate oxidation of $^1\text{O}_2$, the C=C in CDs can be oxidized via π^2 - π^2 cycloaddition to form a high-energetic dioxetane intermediate. (3) This intermediate is can further decompose by spontaneous oxidation to release a lot of chemical energy. (4) The chemical energy can transfer from intermediate to the CDs through electron exchange and further promote the ground-state electrons of the CDs to excited state. (5) The excited CDs then transit to the ground state and produce photon via a radiative recombination. With this continuous photooxidation process, the CDs can produce the long-persistent afterglow.

iii)

L. 273: the fluorescence, absorption spectra and femtosecond transient absorption (TA) spectra of the CDs **before and after light irradiation were detailly characterized. As shown in Figs. Fig. 2g and 2h**, the fluorescence intensity of CDs trends to decrease with the exposure of light irradiation as well as an attenuation in the absorption peak of CDs at 660 nm, declaring that the photophysical properties of CDs also vary with the light irradiation.

L. 305: it can be concluded that the surface of the CDs has underwent ~~ana~~ photo-induced oxidation process as illustrated in Fig. 2o.

L. 447: As shown in Fig. 5a, nearly 100% viability for ~~Hela~~ HeLa cells can be obtained when the concentration of the p-CDs is up to 300 $\mu\text{g mL}^{-2-1}$, **indicating the high biocompatibility of the p-CDs. Secondly, the p-CDs have** been administered orally to mice for 14 days at 0.5 mg mL^{-1} as a therapeutic dose to conduct the safety evaluations at the level of living animals.

L. 555: DFT calculations also confirm that the size-dependent afterglow lifetime of the CDs can be attributed to **the altera–altered activation energy barrier via the steric hindrance effect. By further slowing** down the light-induced-oxidation process via amphipathic molecular coating,

iv) We are very grateful for your careful reading and the detailed suggestion. The SEM image of CDs is less significant for the analysis of CDs. Hence, we did not collect relevant SEM data. We are very sorry about our negligence for the statement of SEM collection and the problems have been corrected in the revised manuscript. In addition, we have added AFM images and the analyzed as follows:

Atomic Force Microscopy (AFM) images have been performed on the CDs-1, CDs-2, and CDs-3, and the results reveal the uniform heights of 8.6, 4.4, and 2.9 nm for the CDs-1, CDs-2 and CDs-3 (Fig. R16), which are consistent with the size variations observed in the TEM images. We have added the above discussion and figures in the revised manuscript and supplementary information.

Fig. R16. The AFM images of CDs-1 **a**, CDs-2 **b**, and CDs-3 **c**.

v) We have annotated the band assignments in the explanation section of the FT-IR in the manuscript (Fig. R17), as follows: As illustrated in Fig. 2d, the FT-IR spectra also reveal the stretching vibration of C–N (1350 cm^{-1}), C=C (1600 cm^{-1}), O–H (3200 cm^{-1}) after the light irradiation.

Sample preparation: The chromatographic samples were incubated at $60\text{ }^{\circ}\text{C}$ in a drying oven for 12 h, and then the solvent was removed to collect the solid-state CDs. The CDs after lasting irradiation were separated with a mixture of methylene chloride and water and the solid-state sample was further collected with a subsequent freeze-drying of the aqueous fraction for the FT-IR measurement.

The FT-IR spectra acquired: the as-prepared solid sample was evenly mixed with potassium bromide at a ratio of 1 : 100. After pressed, the FT-IR spectra of samples were acquired with a Fourier transform infrared spectrometer (Thermo Scientific Nicolet iZ 10). The air background was subtracted when the sample was tested.

We have added the above details in the revised supplementary information.

Fig. R17. Fourier transform infrared spectrum of the CDs before and after light irradiation.

Reviewer #2: Ultralong afterglow emission in organic or inorganic materials usually involves the slow release of trapped charge in a crystal defect or a complicated protection approach for triple-state excitons. However, issues like potential toxicity, poor processability, and serious quenching from oxygen and water still hinder their practical application. The manuscript by Shan and coworkers describes a novel photooxidation strategy to construct extra-long afterglow emissions from carbon nanodots with a lifetime of up to an hour scale. This is a feat in the field of long persistent luminescence. Subsequently, afterglow imaging-guided surgery has been successfully performed to remove the tumor tissues accurately with carbon nanodots, demonstrating their clinical value. We believe this work is meaningful, and well-grounded and will be of indisputable interest to the readership of Nature Communications. Nevertheless, there are still some scientific issues to be clarified. A revision is required to improve this work for publication in this journal. Our comments and questions are listed as follows:

1) CDs in ethanol solutions with different sizes obtain different afterglow lifetimes, while the afterglow lifetime of p-CDs is further improved, but the article does not give the size of p-CDs-1, p-CDs-2, and p-CDs-3, please give the sizes of p-CDs-1, p-CDs-2, and p-CDs-3 respectively, and discuss the reasons for obtaining longer afterglow lifetimes.

Response: Thank you for the valuable suggestion. To probe the size, we have conducted the dynamic light scattering (DLS) tests on p-CDs-1, p-CDs-2, and p-CDs-3. As depicted in the Fig. R18, the dimensions of p-CDs-1, p-CDs-2, and p-CDs-3 are essentially identical at 223.1, 240.6 and 208.6 nm, respectively. In this work, the afterglow of CDs is derived from photooxidation wherein the continuous oxidation of singlet oxygen exerts steric hindrance on CDs, thus facilitating longer lifetimes for larger-sized CDs and achieving an afterglow lifetime of 46.4 min (for CDs-1) in ethanol solution. Furthermore, the surface of CDs undergoes additional passivation through coating treatment with the amphiphilic block copolymer F127, which can further reduce the oxidation rate of singlet oxygen towards p-CDs and result in a

significant increase in the afterglow lifetime.

We have added the above discussion and figures in the revised manuscript.

Fig. R18. Dynamic light scattering (DLS) distribution of the p-CDs-1, p-CDs-2 and p-CDs-3 in aqueous solution.

2) In this paper, the mechanism of photooxidative afterglow is reported, which can effectively avoid the quenching of oxygen, and whether the CDs can achieve afterglow in the solid state, which is very helpful for the discussion of the afterglow mechanism of the CDs, please test the afterglow of the CDs in the solid state and explain it.

Response: Thank you for this valuable question. The possible photooxidative afterglow of the solid-state CDs has been tested. As depicted in the Fig. R19, the CDs in the solid-state exhibit prominent afterglow emission when they are exposed to air. However, when the solid-state CDs are encapsulated in epoxy resin to isolate oxygen, the CDs present no afterglow emission after irradiation (Fig. R20). This result clearly proves that the participation of oxygen is the necessary requirement for the afterglow emission of CDs. Different from phosphorescence or thermally activated delayed fluorescence, the photooxidation afterglow of CDs originates from the continuous oxidation of CDs by singlet oxygen, where oxygen is active involvement in the photooxidation process.

We have added the above discussion and figures in the revised manuscript and supplementary information.

Fig. R19. **a** The photograph of the solid-state CDs under sunlight. **b** The afterglow of the solid-state CDs exposed to air after light irradiation (660 nm, 1.5 W cm^{-2} , 2 min).

Fig. R20. **a** The photograph of the epoxy resin-encapsulated CDs under sunlight. **b** The fluorescence of the epoxy resin-encapsulated CDs light irradiation. **c** The afterglow of the epoxy resin-encapsulated CDs after light irradiation (660 nm, 1.5 W cm^{-2} , 2 min).

3) The biosafety of the CDs was tested by feeding them to mice, but their cytotoxicity seems to be selected only for one kind of tumor cells, the toxicity test of the CDs in different tissue cells should be completed.

Response: Thank you for this helpful comment. The toxicity of CDs in different tumor cell lines was tested, and the results were illustrated in the Fig. R21. At a concentration of 1 mg mL^{-1} p-CDs, the tumor cell lines of MKN-45 exhibit a survival rate of exceeding 90% (Fig. R21a), and the tumor cells can maintain the outstanding viability after 2 and 3 days of cultivation (Fig. R21b and R21c), indicating the

exceptional biocompatibility of p-CDs. Similarly, the p-CDs also present extremely low biological toxicity in the tumor cell lines of KYSE-150 (Fig. R21d-f).

The cytotoxicity observed in previous test may be attributed to potential experimental environment contamination, leading to data inaccuracies. With the multiple loading and repeated experiments, we have gained confidence in the credibility of the test results presented in this study.

We have added the above discussion and figures in the revised manuscript and supplementary information.

Fig. R21. The survival rate of MKN-45 cells with the 1 mg mL^{-1} p-CDs after 1 d **a**, 2 d **b**, and 3 d **c**. The survival rate of KYSE-150 cells with the 1 mg mL^{-1} p-CDs after 1 d **d**, 2 d **e**, and 3 d **f**.

4) The manuscript explains the afterglow lifetime of CDs with different sizes as follows: “Interestingly, compared to small-sized C3 units, larger-sized C5 and GNR units, especially GNR, have higher activation energy in the oxidation process of C=C due to the steric hindrance effects of quantum dots. Thus, as the sample size increases, the energy barrier for oxidation is higher and the oxidation reaction is more difficult to proceed, which means a slower intermediate production rate to further result in a longer afterglow lifetime.” According to this theory, the initial afterglow of C3 should be stronger than that of C5 and GNR, please give the initial afterglow luminous

intensity of the three to verify the correctness of this theory.

Response: We are grateful for the professional suggestion and agree with your opinion. We have compared the change in afterglow intensity of the CDs with different sizes over a 20 min period. The initial afterglow intensity is depicted in the Fig. R22. As previously mentioned, the initial afterglow intensity of CDs-1 (~GNR) is indeed lower than that of CDs-2 (~C5) and CDs-3 (~C3); however, the afterglow decay rate of CDs-1 (~GNR) is significantly slower than that of CDs-2 (~C5) and CDs-3 (~C3). After 10 min, the afterglow intensity of CNR already becomes stronger than that of CDs-2 (~C5) and CDs-3 (~C3). This outcome aligns with the statement has been made in the paper: "Interestingly, when compared to small-sized C3 units, large-sized C5 and GNR units (especially GNR) exhibit higher activation energies during C=C oxidation due to steric hindrance effects caused by quantum dots. Consequently, as sample size increases, so does the energy barrier for oxidation reactions, making it more challenging for intermediate product formation to occur at a slower pace—ultimately leading to an extended afterglow lifetime." The consistency between this theory and the explanation regarding CDs afterglow mechanism validates its accuracy.

Fig. R22. The afterglow intensity of CDs-1 (5.8 nm), CDs-2 (3.4 nm) and CDs-3 (1.9

nm) over a 20 min period after irradiation (660 nm, 1.5 W cm⁻², 1 mg mL⁻¹).

5) The author should provide more evidence on the fluorescence mechanism of CDs, especially its luminescence source at 720 nm.

Response: We are very grateful for your valuable suggestion. To explore the fluorescence mechanism of CDs, the excitation–emission matrix of the CDs was tested. As shown in the Fig. R23a, the CDs present a stable luminescence center with independent excitation wavelength. For the emission of CDs at 670 nm, the absorption and emission spectra of CDs show an ultra-small Stokes shift of 12 nm (Fig. R23b), indicating the interbond exciton recombination and weak electron-phonon coupling interactions. This implies that the emission at 670 nm may originate from the band-edge recombination of CDs.

To further evaluate the intrinsic properties of photo-excitons in CDs, we have plotted a two-dimensional false-color diagram of the temperature-dependent PL spectrum (Fig. R24a). As the temperature increases from 80 to 300 K, the integral PL intensity at 670 nm of CDs decreases to 23%, which indicates high thermal stability for the radiative recombination of the CDs. The corresponding exciton binding energy can be extracted by plotting the integral area of PL emission intensity as a function of temperature according to the following equation:

$$I(T) = \frac{I_0}{1 + A \exp(-E_b / k_B T)} \quad (1)$$

Where I_0 represents the integral intensity of PL emission at 0 K, A is the proportionality constant, E_b is the exciton binding energy, and k_B is the Boltzmann constant. As shown in Fig. R24b, the fitting analysis results show that the CDs have a relatively large E_b of 47.34 meV, indicating their strong exciton recombination ability. High E_b and fast emissivity indicate that the CDs has extraordinary luminescence properties. The emission bandwidth of CDs is considered to be closely related to the strength of electron-phonon coupling. To determine this strength, the Huang-Rhys factor (S) is introduced as a metric to estimate the coupling strength by plotting full width at half maximum (FWHM) as a function of the inverse temperature and fitting

with the following equation:

$$\text{FWHM}(\Gamma) = 2.36\sqrt{S}\hbar\omega_{\text{phonon}}\sqrt{\coth\frac{\hbar\omega_{\text{phonon}}}{2k_{\text{B}}T}} \quad (2)$$

Where S is the yellow-Rhys factor, ω refers to the frequency of phonons, S is estimated to be 0.13 (Fig. R24c). The small S value confirms the weak coupling between electron transitions and lattice phonons in the CDs. In addition, the PL position remains relatively unchanged in the temperature ranging from 80 to 300 K, indicating that the electron-phonon interaction is relatively weak. Based on these results, it can be inferred that the CDs exhibit high structural stiffness and weak electron-phonon interactions, which may contribute to the stable red emission. (Adv. Mater. 2023, 35, 2302275)

For the emission at 720 nm of CDs, there is an approximately 60 nm Stokes shift and a larger FWHM. Additionally, the fluorescence lifetime at 720 nm is measured at 5.43 ns (Fig. S25a), indicating a faster radiative recombination. Moreover, the fluorescence lifetimes at 720 and 670 nm (5.36 ns) exhibit certain differences, suggesting distinct emission sources. According to previous reports (Chem. Sci. 2017, 8, 3599. Adv. Sci. 2022, 2203622. Adv. Mater. 2020, 1906641), it may originate from the surface-localized excitonic vibrational fine emission bands of CDs. This hypothesis can also be confirmed by the spectral changes of the CDs after the surface passivation of F127. As shown in the Fig. R25b, compared with the pure CDs, the surface-passivated p-CDs present a relatively weakened fluorescence emission at 720 nm owing to that the surface passivation of F127 on CDs limits the vibration emission on the surface. Hence, 720 nm related emission is corresponding to the surface-localized excitonic vibrational fine emission bands in the CDs, which is also reflected in the apparent discrepancy presented in the fluorescence and photooxidation afterglow spectra of the CDs (Fig. R25c).

We have added the above discussion and figures in the revised manuscript and supplementary information.

Fig. R23. **a** The excitation–emission matrix of the CDs solution. **b** The UV-Vis absorption and corresponding PL spectrum of CDs.

Fig. R24. **a** The PL emission at 670 nm of the CDs in the temperature range from 80 to 300 K. **b** Integrated PL intensity and **c** FWHM as a function of reciprocal temperature.

Fig. R25. **a** Time resolved decay spectra of the CDs with 670 nm and 720 nm. **b** Fluorescence spectrum of the p-CDs and CDs. **c** Fluorescence and photooxidation afterglow spectra of the CDs.

6) In biological experiments, what is the biological metabolic cycle of the long afterglow CDs? And they can be eliminated from the body through bodily fluids, or not.

Response: Thanks for the helpful comment. According to your suggestion, after injecting p-CDs solution into the tail vein of mice for 2 h, the urine of mice was

collected and tested. As shown in the Fig. R26, obvious fluorescence and afterglow signal can be observed in the urine, indicating that the p-CDs can be excreted normally from the living body after injection. Meanwhile, the mice were dissected after 24 h, and it can be observed that there are only trace residues in the lungs and pancreas except for the strong enrichment of p-CDs in the tumor (Fig. R27). Hence, the CDs can be eliminated from the body through bodily fluids.

Fig. R26. The fluorescence and afterglow imaging of the urine from mice after injecting p-CDs solution into the tail vein of mice for 2 h.

Fig. R27. The fluorescence imaging of CDs in organs of mice after injecting p-CDs solution into the tail vein of mice for 24 h.

We have added the above discussion and figures in the revised manuscript and supplementary information.

7) After hydrophilic treatment, the lifetime of CDs increased by six times. Why is there such a significant improvement in their lifetime? Has the luminescence intensity of CDs of the same size also changed before and after treatment?

Response: Thank you for this valuable question. In this work, the afterglow of the

CDs originates from the continuous oxidation of the CDs by singlet oxygen generated from light irradiation. In this process, the oxidation rate of the CDs can determine the afterglow intensity and luminous duration of the CDs. After the CDs is treated by amphiphilic block polymer polyether F127, the CDs changes from fat soluble to hydrophilic. Furthermore, when the CDs are coated by F127 and the surface is passivated, the oxidation rate of the CDs further reduces, and thus the lifetime of p-CDs is greatly improved. Because the oxidation rate of the CDs decreases after coating, the afterglow intensity of p-CDs is obviously weaker than the initial CDs (Fig. R28). Actually, it can be found that all the afterglow intensities of the p-CDs with the same size decrease significantly after the coating treatment in comparison with the initial CDs.

Fig. R28. The afterglow intensity of different CDs and p-CDs.

8) The reason for selecting the leaves of *Solanum nigrum* L to prepare such carbon dots should be given. I am curious whether other biomass raw materials can achieve similar results.

Response: Thanks for the helpful comment. As reported in previous literatures (Adv. Mater. 2020, 190664. Adv. Mater. 2020, 1906641.), plant leaves are the common raw materials to prepare CDs for bioimaging due to their crucial consideration of

biosafety. *Solanum nigrum* L, as a type of herb and an important Chinese herbal medicine for human diseases therapy, is characterized by high biosafety and wide distribution around the world. Therefore, it is utilized as the raw materials for the synthesis of CDs in our work.

Additionally, we have prepared different products from several plants as the CDs and compared their photophysical properties. As shown in Fig. R29a and R29b, the CDs were obtained from banana peel, carrot, ginger and lettuce. And then, the relevant fluorescence spectra were tested. The CD products prepared from different plants exhibit different fluorescence emission ranging from blue to red. Meanwhile, we have conducted the afterglow emission testing on these CD products. As depicted in Fig. R29c, the products extracted from lettuce has obvious afterglow emission. While, other CDs obtained from banana peels, carrots and ginger did not display any afterglow. These results imply that the CDs may be attributed to the carbonization and polymerization of relevant molecules in specific plants, which have the potential for photooxidation to produce afterglow, but not vice versa.

Fig. R29. The photographs **a**, PL emission **b** and afterglow **c** of CDs derived from

banana peel, carrot, ginger and lettuce leaves.

9) Is the ultralong afterglow emission in the manuscript related to the triplet-state radiative recombination of CDs? The authors should provide other scientific evidence to support this argument.

Response: Thank you for the valuable suggestion. Generally, the phosphorescence or thermally activated delayed fluorescence (TADF) afterglow is originated from triplet-state related recombination. Due to the influence of the excited triplet state of dissolved oxygen, the triplet-state radiative recombination of CDs is highly susceptible and can be quenched in aqueous solutions. Therefore, the most prevalent approach to triplet-state radiative recombination in aqueous solutions is through oxygen isolation. In this study, we have successfully achieved the ultra-long afterglow in aqueous solution by photooxidation and this afterglow accomplishment presents high tolerance for dissolved oxygen. The PL emission and afterglow of the CDs present nearly same spectra (Fig R30), providing evidence that they originate from the same radiative recombination process. Meanwhile, the phosphorescent or TADF emission does not induce the alterations in the structure and optical properties while the CDs will undergo noticeable oxidation and irrecoverable afterglow during the luminescence process in this work. The results conclusively demonstrate that the ultralong afterglow in CDs does not originate from triple excitons. Additionally, compared with the solid-state CDs demonstrating obvious afterglow emission in air, the CDs encapsulated with epoxy resin show almost no afterglow after light irradiation (Fig. R31 and R32). This can clearly verify that the afterglow of CDs requires the participation of oxygen. Different from triplet exciton-related phosphorescence emission, this photooxidation afterglow originates from the continuous oxidation of CDs by singlet oxygen or other ROS species, and oxygen is the active involvement in this oxidation process. In summary, the long-persistent afterglow in CDs here arises from photooxidation effect rather than triple-state radiation recombination.

We have added the above discussion and figures in the revised manuscript and

supplementary information.

Fig. R30. The fluorescence and afterglow spectra of CDs.

Fig. R31. a The photograph of solid-state CDs in air under sunlight. **b** The afterglow imaging of solid-state CDs in air after light irradiation (660 nm , 1.5 W cm^{-2} , 2 min).

Fig. R32. a The photograph of the epoxy resin-encapsulated CDs under sunlight. **b** The fluorescence of the epoxy resin-encapsulated CDs light irradiation. **c** The afterglow of the epoxy resin-encapsulated CDs after light irradiation (660 nm , 1.5 W

cm⁻², 2 min).

Reviewer #3: In this manuscript, the authors report an interesting photooxidation triggered ultralong afterglow, surprisingly, it is possible to achieve an afterglow lifetime of up to 5.9h in carbon nanodots. Conceivably, this approach can be used to the development of new-type afterglow materials without water quenching or toxicity problems. I think this is a solid and important piece of work in the field of long persistent phosphors, which will no doubt have a significant impact for the scientific community of Nature Communications. I recommend publication after addressing the following minor points:

1. In this work, the morphology of CDs-1, CDs-2, and CDs-3 are only investigated with TEM. The high-resolution TEM image and AFM should also be provided, which can further illustrate the exact morphologies of the as-prepared CDs.

Response: Thanks for the helpful comment. As shown in the Fig. R33, the high-resolution TEM images have been incorporated. It can be observed that the CDs-1, CDs-2 and CDs-3 possess the consistent lattice fringes with a spacing of 0.21 nm. Additionally, Atomic Force Microscopy (AFM) images have been performed on the CDs-1, CDs-2, and CDs-3, and the results reveal the uniform heights of 8.6, 4.4, and 2.9 nm for the CDs-1, CDs-2 and CDs-3 (Fig. R34), which are consistent with the size variations observed in the TEM images. We have added the above discussion and figures in the revised manuscript and supplementary information.

Fig. R33. The high-resolution TEM images of CDs-1 **a**, CDs-2 **b**, and CDs-3 **c**.

Fig. R34. The AFM images of CDs-1 **a**, CDs-2 **b**, and CDs-3 **c**.

2. The authors present “obvious afterglow with the wavelength extending to the NIR region can be observed from the CDs, whose afterglow spectra are consistent with their fluorescence spectra (Fig. 1f, 1k and 1p). However, the afterglow spectra of these CDs seem slightly different from the fluorescence. Can the authors provide the exact comparison and discuss the possible reason?”

Response: Thanks for the helpful comment. We have conducted a comparative analysis of the normalized fluorescence and afterglow spectra of CDs. As illustrated in Fig. R35, it can be observed that the fluorescence and afterglow spectra of the CDs are almost same at main peak, while there is only a slight difference in the emission peak at around 720 nm. According to previous reports (Chem. Sci. 2017, 8, 3599. Adv. Sci. 2022, 2203622. Adv. Mater. 2020, 1906641), it may originate from the surface-localized excitonic vibrational fine emission bands of CDs. The influence of surface oxidation on these vibrational fine structures during the afterglow process of CDs may lead to the specific variations at 720 nm.

Fig. R35. The normalized fluorescence and afterglow spectra of CDs.

3. The authors present that “All the afterglow emission time of these CDs in ethanol solution is over 4 h, and the corresponding lifetimes are fitted to 46.4, 42.5 and 29.7 minutes (min) for CDs-1, CDs-2, and CDs-3 (Figs. 1g, 1l and 1q)”. However, the reproducibility of afterglow lifetime has not been provided. Can the authors provide the detailed experimental method and compared the results?

Response: Thanks for the helpful comment. We have conducted the repeated tests on the lifetimes of CDs-1, CDs-2, and CDs-3. As depicted in Fig. R36, R37 and R38, the afterglow decay exhibit almost the same performance, indicating the excellent repeatability of lifetimes for the CDs-1, CDs-2, and CDs-3 (Fig. R39).

Experimental methods: The CDs ethanol solution (2 mL, 1 mg mL⁻¹) was injected into a 5 mL PVC tube. The solution was exposed with 660 nm laser (2 min, 1.5 W cm⁻²). Subsequently, 200 μL of the illuminated CDs solution was transferred to a 96-well black enzyme-linked immunosorbent assay (ELISA) plate. Afterglow testing was conducted using a Bioluminescence mode in an in vivo imaging system with a 670 nm filter, 2 min post-illumination, and 5 min intervals for 4 h.

We have added the above discussion and figures in the revised manuscript and supplementary information.

Fig. R36. Afterglow lifetime curves of the CDs-1.

Fig. R37. Afterglow lifetime curves of the CDs-2.

Fig. R38. Afterglow lifetime curves of the CDs-3.

Fig. R39. The lifetime of CD-1, CD-2 and CD-3.

4. The author claim “photooxidation approach can be used to elucidate the unique afterglow emission”. Whether the common oxidant can trigger the afterglow? This may be a more obvious evidences about the afterglow mechanism.

Response: Thank you for the valuable suggestion. We have compared the afterglow performances of the CDs in common oxidants (active nitrogen, ONOO^- ; hypochlorous acid, ClO^- ; hydrogen peroxide, H_2O_2). As illustrated in Fig. R40, the ONOO^- , ClO^- , and H_2O_2 can trigger the photooxidation afterglow of the CDs. Meanwhile, it can be observed that the afterglow intensity from the CDs increases with the enhanced oxidation, implying that the afterglow of CDs originates from the surface oxidation. We have added the above discussion and figures in the revised manuscript and supplementary information.

Fig. R40. The afterglow performances of CDs in active nitrogen, hypochlorous acid, and hydrogen peroxide and water.

5. The authors present “photooxidation approach can be used to elucidate the unique afterglow emission” and “the energy barrier for oxidation is higher and the oxidation reaction is more difficult to proceed, which means a slower intermediate production rate to further result in a longer afterglow lifetime”. However, the generation capability of $^1\text{O}_2$ is also possible. The authors should provide more evidences to

support the mechanism.

Response: Thanks for the helpful comment. We have conducted a $^1\text{O}_2$ generation test using p-CDs-1, p-CDs-2, and p-CDs-3 under identical conditions. As shown in Fig. R41, there is nearly no significant difference in EPR intensity for the three samples, indicating the similar capability of p-CDs-1, p-CDs-2, and p-CDs-3 to generate $^1\text{O}_2$ under the same light irradiation. Thus, the differences in the afterglow lifetimes among p-CDs-1, p-CDs-2, and p-CDs-3 are likely attributed to variations in oxidation rates induced by the size differences of CDs.

Fig. R41. The EPR signals of p-CDs-1, p-CDs-2 and p-CDs-3 under dark and the same light irradiation.

6. In Fig. 5a, nearly 100% viability for HeLa cells can be obtained when the concentration of the p-CDs is up to 300 ug mL^{-1} . However, when the concentration of the p-CDs reaches 400 ug mL^{-1} , the cell viability seems to be lower than 80%. It is very confusing. The author can provide the IC50 to exactly confirm the high biocompatibility of the p-CDs.

Response: Thank you for this helpful comment. The toxicity of CDs in different tissue cells was tested, and the results were shown in the Fig. R42. At a concentration of 1 mg mL^{-1} , p-CDs exhibited a survival rate consistently exceeding 90% (Fig. R42a), and even after 2 and 3 days of cultivation, the cells maintained outstanding

viability (Fig. R42b and R42c), indicating the exceptional biocompatibility of p-CDs. Additionally, p-CDs demonstrated excellent low biological toxicity across different tissue cells (Fig. R42d to R42f). Furthermore, due to the excellent biocompatibility and high cell survival rate of p-CDs, IC50 has not been calculated.

The cytotoxicity observed in previous test may be attributed to potential experimental environment contamination, leading to data inaccuracies. Through multiple repetitions of the experiments, we have gained confidence in the credibility of the test results presented in this study.

We have added the above discussion and figures in the revised manuscript and supplementary information.

Fig. R42. The toxicity of CDs in different tissue cells.

7. The calculation results are quite interesting. But the description about the calculation is quite confusing. What's the exact code and software for theory calculation in S26? Can the author provide the computational detail, including structural optimization and calculation of transition state potential barriers?

Response: Thank you for your valuable suggestion.

Computational details: The first-principles calculations are based on the density functional theory (DFT) with the projector augmented wave (PAW) method¹ as implemented in the VASP code². The generalized gradient approximation-Perdew-

Burke-Ernzerhof (GGA-PBE) method³ is used to describe the exchange-correlation functions. Dispersion correction of the system is performed using the DFT-D3 method⁴ to better describe the vdW interaction. A plane-wave cutoff energy of 400 eV is used. The Brillouin zones were sampled using k-points with $2\pi \times 0.02 \text{ \AA}^{-1}$ spacing in the Monkhorst–Pack scheme⁵. All atoms are fully relaxed with a tolerance in total energy of 10^{-5} eV, and the forces on each atom are less than 0.02 eV \AA^{-1} . A vacuum of 16 Å perpendiculars to the surface is applied to avoid the interaction between adjacent slabs. The climbing image nudged elastic band (CI-NEB) method⁶ is used to determine the energy barriers of the various kinetic processes.

To analysis the size effects of the carbon nanodots (CDs), three simplified models C3, C5, and graphene nanoribbon (GNR) are used for theoretical calculations. The C3 and C5 contain 3 and 5 six-carbon rings (14 and 22 carbon atoms), respectively. The GNR is modeled in armchair shaped with six-unit cells along the periodic direction (48 carbon atoms). The dangling bonds of carbon atoms at edge are saturated by the H atoms.

References for computational details:

1. Blochl, P. E. Projector augmented-wave method. *Phys. Rev. B* **50**, 17953–17979 (1994).
2. Kresse, G., Furthmüller, J. Efficient iterative schemes for ab initio total-energy calculations using a plane-wave basis set. *Phys. Rev. B* **54**, 11169 (1996).
3. Perdew, J. P., Burke, K. and Ernzerhof, M. Generalized gradient approximation made simple. *Phys. Rev. Lett.* **77**, 3865–3868 (1996).
4. Grimme, S., Antony, J., Ehrlich, S. and Krieg, S. A consistent and accurate ab initio parametrization of density functional dispersion correction (DFT-D) for the 94 elements H-Pu. *J. Chem. Phys.* **132**, 154104 (2010).
5. Monkhorst, H. J., Pack, J. D. Special points for Brillouin-zone integrations. *Phys. Rev. B* **13**, 5188 (1976).
6. Henkelman, G., Uberuaga, B. P., and Jonsson, H. A climbing image nudged elastic band method for finding saddle points and minimum energy paths. *J. Chem. Phys.* **113**, 9901 (2000).

REVIEWER COMMENTS

Reviewer #1 (Remarks to the Author):

The authors have provided a thorough response to the questions arisen by the reviewers and the present version of the manuscript can now be recommended for acceptance.

I have two minor points that may be worth clarifying in the final version of the work:

- 1) Is the production of carbon dots reproducible if a different batch from a different supplier is used?
- 2) Can the authors comment/suggest on a potential strategy to improve the production yield?

Reviewer #2 (Remarks to the Author):

The authors have addressed what I am concerned about. Agree to accept.

Reviewer #3 (Remarks to the Author):

I am satisfied with the authors' response, and the manuscript can be accepted.

Response to Reviewers' comments:

We are very grateful to the reviewers for their thoughtful, detailed and constructive comments on our manuscript (Manuscript ID: NCOMMS-23-42761A). We have checked the comments carefully and revised the manuscript according to these comments. Revised portion is marked in red in the manuscript and Supplementary Information. The point-by-point response to the comments is highlighted in blue style and listed as follows:

Reviewer #1 (Remarks to the Author):

The authors have provided a thorough response to the questions arisen by the reviewers and the present version of the manuscript can now be recommended for acceptance.

Response: We feel grateful to you for your encouragement and positive comments on our work.

I have two minor points that may be worth clarifying in the final version of the work:

1) Is the production of carbon dots reproducible if a different batch from a different supplier is used?

Response: Thank you for the helpful suggestions. Accordingly, the plant sources (*Solanum nigrum* L) for the production of carbon dots (CDs) were purchased from different suppliers, including Bozhou Kangyiyin Biotechnology Co., Ltd., Henan Green He Pharmaceutical Co., Ltd., and Bozhou Haoyitang Biotechnology Co., Ltd. The afterglow CDs were prepared with the plant sources from the three suppliers under the same condition of solvothermal and chromatographic treatment (Fig. R1a, R1f and R1k). The potential variations of these CDs were investigated. As illustrated in Fig. R1b, R1c, R1g, R1h, R1l, and R1m, these batches of CDs from different suppliers were checked with fluorescence and absorption spectra. And the results revealed that these CDs possessed almost the same fluorescence and UV-vis absorption spectra without obvious peak shift. Meanwhile, the PL QYs of these batches of CDs also demonstrated the similar PL QYs of 28.42%, 27.86% and 28.44% for these CDs (Fig. R1d, R1i and R1n), indicating the excellent repeatability of the

preparation approach to the CDs from batch to batch. In addition, the production yields of CDs prepared from different suppliers were also investigated. As shown in the Fig. R1e, R1j and R1o, the production yields of 1.3%, 0.8% and 1.8% could be obtained from different suppliers. The above results clearly indicated that the production of CDs is reproducible if a different batch from a different supplier is used while only there is difference in production yields.

We have added the above contents and figures in the revised supplementary information.

Fig. R1. **a-e** **a** The photograph of *Solanum nigrum* L and **b** fluorescence spectrum, **c** absorption spectrum, **d** PLQY and **e** production yield of the CDs prepared from the source purchased from Bozhou Kangyiyin Biotechnology Co., Ltd. **f-j** **f** The photograph of *Solanum nigrum* L and **g** fluorescence spectrum, **h** absorption spectrum, **i** PLQY and **j** production yield of the CDs prepared from the source purchased from Henan Green He Pharmaceutical Co., Ltd. **k-o** **k** The photograph of *Solanum nigrum* L and **l** fluorescence spectrum, **m** absorption spectrum, **n** PL QY and **o** production yield of the CDs prepared from the source purchased from Bozhou Haoyitang Biotechnology Co., Ltd.

2) Can the authors comment/suggest on a potential strategy to improve the production yield?

Response: Thank you for the valuable comment. Generally, industrial production often involves using larger high-pressure vessels to expand the reaction and yield. Thus, we have compared the production yield of CDs using the poly (tetrafluoroethylene) (Teflon)-lined autoclaves with different volumes. As shown in the Fig. R2a, 20, 50 and 100 mL autoclaves were used to prepare the CDs with the same source of 1 g plants and 10, 20 and 50 mL ethanol, respectively. As a result, the weights of the obtained CDs finally were 0.009, 0.0133 and 0.0287 g, and the corresponding production yields were 0.9%, 1.3% and 2.9%, respectively (Fig. R2b to R2d). The results clearly indicate that large reaction chamber can effectively improve the production yields of CDs, which may be attributed to the solvent to raw material contact ratio and the shift of reaction balance. We believe the potential strategy to improve the production yield of CDs by enlarging the reaction chamber is also suitable for the large-scale industrial production.

Fig. R2. **a** The photograph of the poly (tetrafluoroethylene) (Teflon)-lined autoclaves with different volumes. **b** Weights of the initial flask and the flask with the CD products from the 25 mL reactors. **c** Weights of the initial flask and the flask with the CD products from the 50 mL reactors. **d** Weights of the initial flask and the flask with the CD products from the 100 mL reactors.

Reviewer #2 (Remarks to the Author):

The authors have addressed what I am concerned about. Agree to accept.

Response: We sincerely appreciate you for your strong approval and kind encouragement.

Reviewer #3 (Remarks to the Author):

I am satisfied with the authors' response, and the manuscript can be accepted.

Response: Thank you very much for your support and affirmation.

REVIEWERS' COMMENTS

Reviewer #1 (Remarks to the Author):

The authors have addressed the minor questions properly.

Response to Reviewers' comments:

We are very grateful to the reviewers for their thoughtful, detailed and constructive comments on our manuscript (Manuscript ID: NCOMMS-23-42761B). We have checked the comments carefully and revised the manuscript according to these comments. Revised portion is marked in **red** in the manuscript and Supplementary Information. The point-by-point response to the comments is highlighted in **blue style** and listed as follows:

Reviewer #1 (Remarks to the Author):

The authors have addressed the minor questions properly.

Response: We sincerely appreciate you for your strong approval and kind encouragement.